# Token Distillation: Attention-Aware Input Embeddings for New Tokens

**Konstantin Dobler**
Hasso Plattner Institute
ELLIS Unit Potsdam
konstantin.dobler@hpi.de

**Desmond Elliott**
University of Copenhagen
ELLIS Unit Copenhagen
de@di.ku.dk

**Gerard de Melo**
Hasso Plattner Institute
ELLIS Unit Potsdam
gerard.demelo@hpi.de

## Abstract

Current language models rely on static vocabularies determined at pretraining time, which can lead to decreased performance and increased computational cost for domains underrepresented in the original vocabulary. New tokens can be added to solve this problem, when coupled with a good initialization for their new embeddings. However, existing embedding initialization methods require expensive further training or pretraining of additional modules. In this paper, we propose Token Distillation and show that by distilling representations obtained using the original tokenization, we can quickly learn high-quality input embeddings for new tokens. Experimental results with a wide range of open-weight models show that Token Distillation outperforms even strong baselines.[1]

## 1 Introduction

Pretrained language models are trained with a fixed tokenizer that often fragments domain-specific or novel terms into multiple subtokens. This excessive tokenization not only leads to reduced performance on downstream tasks (Rust et al., 2021; Ali et al., 2024) but also increases the computational (and therefore also financial) cost due to inflated sequence lengths (Ahia et al., 2023; Yamaguchi et al., 2024a). Although adding new tokens to a model's vocabulary can reduce over-tokenization, it is crucial to choose a good initialization for the new embeddings (Gee et al., 2022; Minixhofer et al., 2022; Dobler & de Melo, 2023; Yamaguchi et al., 2024a).

In this paper, we argue that many recent methods for embedding initialization are fundamentally limited. Whenever we wish to add a new token to a pretrained model's vocabulary, this new token may be split up into multiple *subtokens* in the original model's tokenization. However, these subtokens might not be individually informative about the semantics of the entire new token (consider, *e.g.*, `<_pal> <at> <able>`). The semantics of a word composed of multiple subtokens will largely not be stored in their raw input embeddings at all – but rather constructed by the Transformer's attention/feed-forward layer stack during contextualization (Elhage et al., 2022; Lad et al., 2024; Kaplan et al., 2025).

Therefore, methods that do not exploit the information encoded in Transformer layer weights are at a serious disadvantage. Motivated by this insight, we propose a novel method for input embedding initialization that captures information stored in all Transformer layers in addition to the existing input embeddings. Our method optimizes new token embeddings with a distillation-based objective to match the model's behavior compared to when a new token is split up into its original subtokens. We demonstrate the efficacy of our method, dubbed "Token Distillation", in Section 5. We illustrate our method in Figure 1 and describe it in Section 3.

We compare against strong baselines, including standard embedding initialization procedures, training of the embedding matrices with causal language modeling, as well as pretrained hyper-networks that predict new token embeddings. Extensive experiments on a wide range of open-weight models, including question-answering benchmarks and the generation of definitions for the newly added tokens confirm the effectiveness of our approach. Our experimental setup is detailed in Section 4. Additionally, we describe related work in Section 2 and explicitly discuss limitations of our method in Appendix A. In summary, our contributions are as follows.

- We propose Token Distillation, a novel method for providing high-quality input embeddings when adding new tokens to pretrained language models.

---

[1]Our code is available at https://github.com/konstantinjdobler/token-distillation.

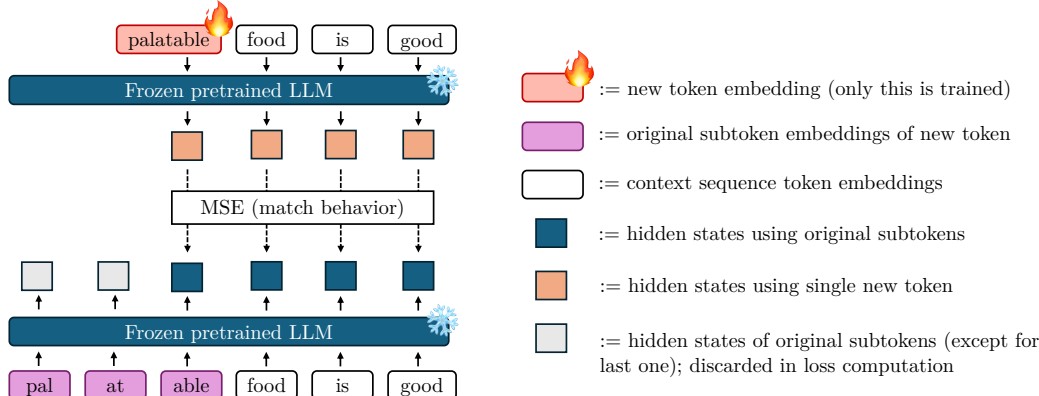

Figure 1: Illustration of Token Distillation – Given a sequence containing our new target token, we first obtain the model's hidden states on that sequence using the original tokenization and then quickly learn a new embedding by reducing the mean squared error (MSE) between the original hidden states and the hidden states of the model when using a single token embedding to replace the original subtokens.

- We motivate our proposed method by describing the fundamental limitations of current embedding initialization methods and empirically verify our claims.
- Extensive experiments and in-depth analyses of our method using a wide range of open-weight model checkpoints and strong baselines confirm its effectiveness.

## 2 BACKGROUND

Most state-of-the-art Large Language Models (LLMs) are trained using a static tokenizer, usually derived by a byte-pair encoding scheme before model training (Sennrich et al., 2016). However, a single tokenizer predetermined before model training might not be equally well-suited for the many different use cases that a model will need to address later on. Suboptimal tokenization of new languages or domain-specific text increases the cost of further training and inference (Ahia et al., 2023; Yamaguchi et al., 2024a) and has also been tied to reduced performance (Rust et al., 2021; Ali et al., 2024). Furthermore, Lesci et al. (2025) show that in practice, words which are not a single token in the vocabulary are significantly less likely to be generated.

A solution to this problem is to modify the existing vocabulary to suit the specific needs. However, simply modifying the vocabulary is insufficient – we also need suitable embeddings for the new tokens. A common approach is simple random initialization followed by a phase of training only the new embedding layers on the target corpus to "warm up" the new embedding weights. However, random initialization of new token embeddings has been widely shown to underperform better methods (*e.g.*, Gee et al., 2022; Minixhofer et al., 2022; Dobler & de Melo, 2023). This has led to notable interest in devising ways of obtaining more informative embeddings for new tokens.

A common alternative is to initialize a new token's embedding as the mean of its subtoken embeddings (Sachidananda et al., 2021; Koto et al., 2021; Gee et al., 2022) – potentially weighted based on the subtoken position in the new token (Nakash et al., 2025) – or based on other heuristics (Downey et al., 2023). Alternatively, a weighted mean of existing embeddings can also be computed based on similarity scores in auxiliary embedding spaces (Wang et al., 2019; Tran, 2020; Minixhofer et al., 2022; Ostendorff & Rehm, 2023; Dobler & de Melo, 2023; Liu et al., 2024, *inter alia*). In fact, most recent research into embedding initialization for new tokens proposes some variation of a weighted linear combination or copy of existing embedding vectors (Mosin et al., 2023; Zeng et al., 2023; Mundra et al., 2024; Yamaguchi et al., 2024a;b; Remy et al., 2024; Goddard & Neto, 2025; Lee et al., 2025; Singh et al., 2025). Nonetheless, the resulting embeddings still require further tuning to achieve good results (Minixhofer et al., 2024). We refer to this category of initialization methods as "model-gradient free", as they do not consider any training signal from the model.

Only a few methods based on access to model gradients have been investigated in recent years. Model gradients can be used by directly optimizing just the new token embeddings to minimize a cross-entropy loss on relevant contexts (Lampinen & McClelland, 2018), the common practice of

training all embeddings while freezing other parts of the model (Artetxe et al., 2020; de Vries & Nissim, 2021) – potentially only backpropagating through parts of the model (Marchisio et al., 2023) – or via (pre-)training a hyper-network (Ha et al., 2017) that is able to predict embeddings for new tokens in the target embedding space (Pinter et al., 2017; Schick & Schütze, 2019; 2020; Teehan et al., 2024; Minixhofer et al., 2024). Alternatively, using PatchScopes (Ghandeharioun et al., 2024), "Tokens to Words" (Kaplan et al., 2025) locates layers at which a word fragmented into multiple subtokens is represented by a single contextual embedding in the residual stream, and then projects these hidden states into the input/output embedding spaces via trained mappings. These methods – hyper-networks in particular – have been shown to yield immediately useful representations but require additional compute for (pre-)training. In the case of hyper-networks, it additionally might be necessary to train different hyper-networks for different target domains, further increasing the complexity.[2] Motivated by these difficulties, we propose an alternative method in this paper that does not require extensive pretraining of an embedding prediction hyper-network or additional modules and still achieves competitive results, even outperforming pretrained hyper-networks.

## 3  METHOD: TOKEN DISTILLATION

Our goal is to initialize input embeddings for new tokens such that, when inserted into a frozen pretrained Transformer, they faithfully reproduce its original behavior – now using a single new token instead of multiple subtokens. Specifically, we seek an embedding $\mathbf{e}^\star$ for a new token $t^\star$ such that all downstream hidden states match those obtained when the model would instead have seen the original subtokens $[t_1, \ldots, t_n]$, which $t^\star$ would have been tokenized into according to the original vocabulary. In what follows, we (1) explain why existing methods fall short and motivate our method, (2) formalize our method, and (3) describe implementation details.

### 3.1  INTUITION

As illustrated in Section 1, individual embeddings for subtokens $t_1, \ldots, t_n$ (*e.g.*, `<_pal> <at> <able>`) of a new token $t^\star$ (*e.g.*, `<_palatable>`) do not necessarily store the semantics of the specific concatenated form $t^\star$. Instead, their contexualized representation is gradually constructed in the Transformer layers via a neural *detokenization* process (Elhage et al., 2022; Lad et al., 2024; Kaplan et al., 2025) and then attended to by other token positions in the sequence, which are influenced by this contextualized representation. The various prior approaches that solely rely on information encoded in the embedding matrices of a model thus ignore the crucial functional knowledge stored in the attention and feed-forward weights of Transformer layers, which imposes a fundamental limitation on their performance.

Precisely pinpointing the specifically involved attention heads is difficult and can require manual inspection (Elhage et al., 2022). Instead, we propose to *distill* (Hinton et al., 2015; Snell et al., 2022) the impact that the multiple subtokens $t_1, \ldots, t_n$ have on other tokens attending to them into a single token embedding $\mathbf{e}^\star$. Our intuition is as follows: If we identify an embedding $\mathbf{e}^\star$ for $t^\star$ such that the model produces similar hidden states in the succeeding positions after seeing $t^\star$ and $t^\star$'s original subtokens $t_1, \ldots, t_n$, we have extracted the relevant information stored in the model's Transformer layers without requiring a specific localization within the model weights. Thus, we propose to optimize $\mathbf{e}^\star$ by minimizing the mean-squared error between hidden states produced by the input sequences containing $t_1, \ldots, t_n$ and their counterparts using $t^\star$. We will show in Section 5 that this empirically outperforms existing training-free embedding initialization methods, embedding tuning with a next-token prediction objective, as well as pretrained embedding prediction hyper-networks. Next, we formally describe our method.

### 3.2  METHOD DESCRIPTION

Let $\tau$ be the tokenizer of a pretrained Transformer. Given a new token string $t^\star$ with original subtokens $\tau(t^\star) = [t_1, \ldots, t_n]$, we require a small corpus of example sequences $s \in \mathcal{S}$ each containing the new token string $t^\star$. We denote the new tokenizer, which includes $t^\star$, as $\tau^\star$, and the tokenization of $s$ using $\tau$ or $\tau^\star$ as $s_\tau$ and $s_{\tau^\star}$, respectively. The target for learning the embedding of $t^\star$ are the hidden states at a specified layer $l$ produced by the language model when processing $s_\tau$, which we denote as $\mathcal{H}^{(l)}(s_\tau)$. We wish to find an embedding $\mathbf{e}^\star$ for a new token $t^\star$ such that hidden states

---

[2]Minixhofer et al. (2024) train separate hyper-networks for `mistralai/Mistral-7B-v0.1` for English+Code and multilingual target tokenizers.

produced by processing the input sequence $s_{\tau^\star}$ instead of $s_\tau$ are similar to our target $\mathcal{H}^{(l)}(s_\tau)$. We denote these new hidden states using our new embedding $\mathbf{e}^\star$ as $\mathcal{H}^{(l)}_{\mathbf{e}^\star}(s_{\tau^\star})$. As the two input sequence indices are not aligned due to replacing subtokens by a single new token, we define a set of mapped positions $(i, j) \in \mathcal{M}(s_\tau, s_{\tau^\star})$, where $i$ and $j$ are the positions of the same token in $s_{\tau^\star}$ and $s_\tau$, respectively. Additionally, $\mathcal{M}(s_\tau, s_{\tau^\star})$ only includes pairs $(i, j)$ where position $i$ in $s_{\tau^\star}$ would attend to $t^\star$. We use subscripts such as $\mathcal{H}^{(l)}(s_\tau)_j$ to signify the hidden state at the $j$-th token position. We learn $\mathbf{e}^\star$ by minimizing the mean-squared error between hidden states for a given target layer $l$:

$$\min_{\mathbf{e}^\star \in \mathbb{R}^d} \mathbb{E}_{s \sim S} \left[ \frac{1}{|\mathcal{M}(s_\tau, s_{\tau^\star})|} \sum_{(i,j) \in \mathcal{M}(s_\tau, s_{\tau^\star})} \left\| \mathcal{H}^{(l)}_{\mathbf{e}^\star}(s_{\tau^\star})_i - \mathcal{H}^{(l)}(s_\tau)_j \right\|_2^2 \right]. \tag{1}$$

In practice, we simply use the last layer's hidden state but analyze this choice in Section 5.3.

### 3.3 FURTHER DETAILS

**Retrieving relevant contexts for new tokens.** Like other methods optimizing embeddings based on contexts, our method also needs input sequences. Firstly, we note that randomly sampling texts from a domain-specific or general corpus is inefficient for our goal of learning newly added embeddings: Typically, the new tokens will only make up a small fraction, so most gradient updates will actually not affect the new input embeddings at all and simply learn to minimize their log-probability in the case of output embeddings. As we aim to have a fast method, we need a better approach. We propose two different approaches: (1) Our main approach is to simply retrieve snippets that contain our target tokens from a domain-specific or general corpus. This can be implemented efficiently using the algorithm proposed by Aho & Corasick (1975). Then we can truncate the snippets to a small window around our target token to optimize computational efficiency. (2) For causal language models, we can simply generate snippets by prompting the model with our target token. We provide implementation details in Appendix C.8. In our main experiments, we focus on the first approach but note the availability of the second approach in case a reference corpus is not available. In most cases of domain and language adaptation, the availability of such a corpus is a reasonable assumption. Nevertheless, we study an ablation instead using the generative approach in Section 5.3.

**Output embeddings.** Since our method backpropagates gradients back from the hidden states, we do not learn output embeddings with our distillation-based objective. In fact, this is not possible, as our new tokens are not part of the original model that serves as the "teacher". In practice, for learning output embeddings, we can simply add a next-token prediction objective just for the output embeddings at a minimal computational overhead or freely combine our method with any other method for initializing output embeddings. Recent work suggests that input and output embeddings should in fact be treated differently (Nakash et al., 2025; Huang et al., 2025). Indeed, the Transformer architecture can handle tokens that are input-only. Therefore, if not paired with an additional initialization method for output embeddings, we have the choice of either adding new tokens only to the input embedding matrix or – for compatibility with existing frameworks – setting their output embeddings to a vector of zeros.

**Hyperparameters.** We employ a simple setup and use the AdamW (Kingma & Ba, 2017; Loshchilov & Hutter, 2019) optimizer for all trainable parameters. We set the batch size to optimize throughput and run all experiments on Nvidia H100 80GB GPUs. We do not use weight decay and maintain a constant learning rate with a linear warmup. For fair comparison, we sweep for the best learning rate for all methods that require a learning rate. Since our method is aimed to serve as an *initialization* rather than as full-scale further training, we restrict the number of example sequences to a maximum of 25 per target token and truncate to a context length of 50 tokens. These restrictions ensure that our method is quick to run, initializing 2,500 new tokens on a single GPU in under 10 minutes. In Section 5.2, we additionally report experiments with a larger compute budget (100 snippets per token) and with continued training for vocabulary adaptation. We use the same data for all training-based methods, including baselines and variations of our method. We provide all details in Appendix C.

## 4 EXPERIMENTAL SETUP

### 4.1 EVALUATION

Common use cases for adding new tokens to a pretrained model's tokenizer are language and domain adaptation, especially for the biomedical domain (Poerner et al., 2020; Gee et al., 2023; Hasan et al., 2024; Singh et al., 2024; Balde et al., 2024). This is because it is a particularly challenging domain with highly complex domain-specific terminology, which can serve as a benchmark to stress test embedding initialization methods. Therefore, we evaluate our method on a collection of standard benchmarks in the biomedical domain (Pal et al., 2024) and add frequently occurring words as new tokens. Additionally, for a more in-depth evaluation of the quality of new representations provided by our methods, we prompt the adapted models to generate definitions for such new tokens and evaluate their quality with an LLM-as-a-Judge (Li et al., 2023). We also evaluate Token Distillation for language adaptation, choosing French as an exemplary language. We evaluate our method on a set of multiple-choice benchmarks from FrenchBench (Faysse et al., 2025) and add frequently occurring words as new tokens, as in the biomedical domain adaptation experiments.

Furthermore, to push the limits of embedding initialization methods, we move beyond domain and language adaptation and apply our method to multi-word tokens, which have recently gained further interest (Gee et al., 2023; Liu et al., 2025; Huang et al., 2025). We follow the zero-shot tokenizer transfer setting in our evaluations (Minixhofer et al., 2024), *i.e.*, we evaluate initialization methods without further training on a corpus, as we want to directly judge the quality of the resulting representations. In Section 5.2, we additionally experiment with a short continued training phase after initializing new embeddings.

We select target tokens that actually occur frequently in the chosen benchmarks so that we can effectively judge the quality of newly generated embeddings. We select words that occur more frequently than a threshold and additionally ensure that all individual benchmarks are represented. We report the full methodology in Appendix C.1. For our multi-word token experiments, we prompt `gpt-o4-mini-high` to generate suitable candidates (full details in Appendix C.3). For our experiments in the biomedical domain, we retrieve contexts from `ncbi/pubmed` as a reference corpus. For our experiments on language adaptation to French, we use `HuggingFaceFW/fineweb-2`. For our multi-word token experiments, we prompt the original models to generate sequences containing the new tokens. For the LLM-as-a-Judge evaluations, we use `Llama3.3-70B-Instruct` as the judge model. We evaluate the general *correctness* of the generated definitions to test the quality and completeness of the resulting representations as well as *semantic similarity* with the target model's output (Villegas et al., 2025), as inducing as little behavior change as possible can be an important desideratum.

### 4.2 MODELING

**Considered base models.** We conduct experiments using a wide range of open-weight model checkpoints to ensure our results are not merely a function of peculiarities in any specific model's embedding space. In particular, we choose the following models: `Mistral-7B-v0.1`, `OLMo2-7B-1124-Instruct`, `Llama3-8B`, `Llama3-8B-Instruct`, `Llama3.1-8B`, `Llama3.1-8B-Instruct`, `Llama3.2-3B`, and `Llama3.2-3B-Instruct`. In the remainder of the paper, we denote "`-Instruct`" variants with a simple "`-i`" suffix. Through this extensive evaluation, we study different model families, model sizes (`3B`, `7B`, and `8B`), instruction-tuned and base models, as well as models with separate input/output embeddings and tied (shared) embeddings between input and output. For our experiments on French language adaptation, we run experiments with the `Mistral-7B-v0.1` and `Llama3-8B(-Instruct)` models and additionally consider `Qwen3-8B-Base`, which has received more extensive multilingual training.

**Baselines.** For all results, we report the performance of the unmodified base model using the original tokenization. To establish a lower bound, we also report results for initializing the new token embeddings randomly from a normal distribution with a per-channel mean and standard deviation of the original embeddings (Hewitt, 2021). Furthermore, we report results for the commonly used method of taking the subtoken mean (Sachidananda et al., 2021; Koto et al., 2021; Gee et al., 2022) as initialization for a new token, which has been shown to perform similarly to more sophisticated initialization methods that also use a weighted average of existing embeddings (Minixhofer et al., 2024; Yamaguchi et al., 2024b).

| | Mistral-7B | Llama3-8B | Llama3-8B-i | Llama3.1-8B | Llama3.1-8B-i | Llama3.2-3B | Llama3.2-3B-i | OLMo2-7B-i | Avg. |
|---|---|---|---|---|---|---|---|---|---|
| Original tokenization | 64.5 | 69.8 | 70.6 | 69.2 | 72.3 | 60.1 | 64.4 | 61.3 | 66.5 |
| Random (Hewitt, 2021) | 57.0±0.6 | 58.8±0.5 | 60.6±0.3 | 59.3±0.5 | 62.6±0.6 | 51.6±0.4 | 55.8±0.2 | 53.9±0.6 | 57.5 |
| NTP (*tune all embeddings*) | 55.7±0.5 | 63.8±0.1 | 63.8±0.2 | 62.9±0.4 | 66.8±0.4 | 51.5±0.6 | 55.5±0.5 | 58.0±0.3 | 59.8 |
| Mean (Gee et al., 2022) | 58.3 | 63.8 | 63.4 | 63.7 | 66.9 | 54.2 | 58.4 | 58.0 | 60.8 |
| NTP (Lampinen et al., 2018) | 61.2±0.4 | 65.6±0.3 | 65.3±0.6 | 66.0±0.5 | 68.9±0.2 | 56.6±0.6 | 61.3±0.5 | 58.7±0.6 | 63.0 |
| ZeTT (Minixhofer et al., 2024) | 62.7 | 66.1 | 66.3 | – | – | – | – | – | – |
| ⋆Token Distillation | 62.8±0.5 | 67.3±0.2 | 67.6±0.3 | 67.3±0.5 | **71.0±0.2** | 56.2±1.9 | **63.1±0.2** | 61.2±0.3 | 64.6 |
| ⋆Token Distillation + NTP | 63.0±0.5 | 67.2±0.3 | 66.7±1.6 | 67.2±0.3 | 70.6±0.4 | 57.6±0.3 | 62.2±0.3 | 59.8±0.5 | 64.3 |
| ⋆Token Distillation + αNTP | 62.8±0.5 | **67.6±0.4** | **67.8±0.5** | 67.4±0.4 | 70.9±0.3 | **57.9±0.1** | 62.5±0.6 | 61.2±0.2 | **64.7** |

Table 1: Benchmark results on biomedical domain adaptation for different initialization methods. We report a macro-average ± standard deviation over five random seeds on the tasks in the Open Medical-LLM leaderboard (see Section 4.1). The best initialization result for each model is given in **boldface**, while all results that are not significantly worse (one-sided Welch's $t$-test with Bonferroni correction, $p < 0.05$) are underlined. Methods without ±x.x are deterministic. −: We only report results for ZeTT where pretrained hyper-networks are available. ⋆: Our method(s).

| Method | Mistral-7B | | Llama3-8B | | Llama3-8B-i | | Llama3.1-8B | | Llama3.1-8B-i | | Llama3.2-3B | | Llama3.2-3B-i | | OLMo2-7B-i | | Avg. | |
|---|---|---|---|---|---|---|---|---|---|---|---|---|---|---|---|---|---|---|
| | Sim | Corr | Sim | Corr | Sim | Corr | Sim | Corr | Sim | Corr | Sim | Corr | Sim | Corr | Sim | Corr | Sim | Corr |
| Original tokenization | 99.8 | 96.5 | 100.0 | 94.3 | 100.0 | 98.4 | 100.0 | 94.7 | 100.0 | 98.4 | 100.0 | 93.2 | 100.0 | 93.8 | 99.4 | 96.9 | 99.9 | 95.8 |
| Random (Hewitt, 2021) | 0.0 | 0.4 | 0.0 | 0.0 | 0.0 | 0.2 | 0.0 | 0.0 | 0.0 | 0.0 | 0.0 | 0.0 | 0.0 | 0.0 | 0.0 | 0.0 | 0.0 | 0.1 |
| Mean (Gee et al., 2022) | 2.5 | 4.3 | 16.8 | 16.2 | 17.6 | 20.7 | 25.6 | 25.6 | 23.2 | 28.3 | 15.2 | 15.2 | 12.1 | 14.6 | 20.1 | 24.2 | 16.6 | 18.6 |
| NTP (*tune all embeddings*) | 26.8 | 30.3 | 33.2 | 39.5 | 34.4 | 43.4 | 28.5 | 35.0 | 41.4 | 50.0 | 19.7 | 20.7 | 18.6 | 21.1 | 22.5 | 26.6 | 28.1 | 33.3 |
| NTP (Lampinen et al., 2018) | 49.8 | 58.2 | 45.3 | 49.6 | 48.4 | 58.0 | 58.6 | 65.6 | 60.2 | 69.9 | 51.6 | 58.4 | 50.0 | 58.8 | 52.1 | 57.0 | 52.0 | 59.4 |
| ZeTT (Minixhofer et al., 2024) | 63.5 | 68.6 | 69.5 | 73.6 | 70.7 | 80.3 | – | | – | | – | | – | | – | | – | |
| ⋆Token Distillation | **79.7** | **85.2** | **72.7** | **76.2** | 79.7 | 91.0 | 75.8 | 80.1 | **81.6** | **89.8** | 0.0 | 0.0 | **75.0** | **83.0** | 83.8 | 89.5 | 68.5 | 74.4 |
| ⋆Token Distillation + NTP | 77.3 | 82.6 | 68.4 | 73.8 | 75.8 | 86.3 | 72.3 | 79.9 | 77.3 | 86.1 | 62.3 | 69.1 | 65.2 | 75.8 | 65.4 | 74.6 | 70.5 | 78.5 |
| ⋆Token Distillation + αNTP | 76.8 | 82.0 | 71.7 | 75.2 | **80.9** | **91.6** | **77.5** | **81.2** | 81.1 | 89.5 | **72.7** | **79.5** | 71.9 | 80.5 | 81.2 | 86.9 | **76.7** | **83.3** |

Table 2: Results for prompting models to generate definitions for newly initialized biomedical domain tokens. We report similarity with the original target model's definition (Sim) and correctness (Corr), both judged by `Llama3.3-70B-Instruct`. We **bold** the best initialization result in each column. ⋆: Our method(s). For full judge prompting details, see Appendix C.7.

Since our proposed method conducts a short optimization on a few reference sequences per new token, we compare against the common approach of training embeddings using causal language modeling using the same data. Specifically, we report results for "classic" embedding tuning using the next-token prediction objective – denoted as NTP (*tune all embeddings*) – as well as masking updates to the original embeddings such that only new embeddings are optimized (denoted as just NTP), which corresponds to the method used by Lampinen & McClelland (2018). These methods use the subtoken mean as their starting point. A strong alternative method for obtaining embeddings for new tokens are hyper-networks specifically pretrained for this task. We report results for ZeTT (Minixhofer et al., 2024) using their provided hyper-network checkpoints.

**Our method.** As the initial embedding input into our method, we also use the subtoken mean. We investigate alternatives to this choice in Section 5.3. Our proposed objective can also be combined with the NTP-based objectives. We report results for combining Token Distillation and NTP. Since the NTP and Token Distillation objectives have different scales, a simple addition is suboptimal. In fact, NTP is usually of larger magnitude, which leads it to overpower Token Distillation. Therefore, we also consider an "autoscaled" variant Token Distillation + αNTP where the NTP loss is scaled by $\alpha$ = stop_gradient(loss_token_distillation / loss_ntp) before summing. Here, stop_gradient prevents gradient flow through the scaling factor $\alpha$.[3]

## 5 RESULTS

### 5.1 MAIN EXPERIMENTS

**Main results.** We provide our main results on benchmarks in Table 1 (biomedical domain adaptation) and Table 3 (French language adaptation). In Table 2, we report our experiments on definition generation for newly added biomedical tokens (reporting similarity to the original target model's generation and correctness as judged by a larger LLM). On average, our method outperforms all other baseline embedding initialization methods. In particular, Token Distillation shows superior results on benchmarks and in its ability to generate correct definitions, which is indicative of higher-quality

---

[3]Multi-objective optimization is a rich field with extensive literature. We leave a thorough investigation of alternatives such as GradNorm (Chen et al., 2018) for future work.

| | Mistral-7B | Llama3-8B | Llama3-8B-i | Qwen3-8B | Avg. |
|---|---|---|---|---|---|
| Original tokenization | 69.5 | 69.4 | 72.1 | 81.7 | 73.2 |
| Random (Hewitt, 2021) | 51.0 | 46.1 | 51.5 | 62.1 | 52.7 |
| Mean (Gee et al., 2022) | 56.3 | 58.4 | 61.7 | 69.6 | 61.5 |
| NTP (Lampinen et al., 2018) | 64.7 | 67.0 | 70.1 | 81.2 | 70.8 |
| ZeTT (Minixhofer et al., 2024) | 66.1 | 61.3[†] | 65.2[†] | – | – |
| ⋆ Token Distillation | **68.5** | **68.9** | **72.9** | **81.5** | **72.9** |

Table 3: Performance on FrenchBench multiple-choice tasks after adding new French whole-word tokens. [†]: For ZeTT, only Mistral-7B has a multilingual-tuned hyper-network, while Llama3-8B(-i) uses the general English+Code hyper-network.

representations. Additionally, Token Distillation also exhibits greater similarity to the original tokenization behavior than all other methods, which substantiates our claim that a distillation-based objective is able to better approximate complex multi-subtoken interactions into a single token embedding.

Note that even with its very competitive performance, Token Distillation does not fully attain the level of the original tokenization, which benefits from the massive pretraining of the original model. This is expected in our zero-shot tokenizer transfer setting without any further training (see Minixhofer et al., 2024), although remarkably, Token Distillation is able to outperform the original tokenization when adding French tokens to `Llama3-8B-i` *without any training* of the Transformer layers. We seek to directly investigate the embedding initialization quality and the advantages of our distillation-based objective compared to next-token prediction rather than the effects of further training. However, results can naturally be improved via further training; we argue that Token Distillation will provide the best starting point for such further processing. In Section 5.2, we additionally experiment with giving Token Distillation additional computational budget as well as a moderate continued training phase, which does enable Token Distillation to closely match the original tokenization's performance while offering significant speedups due to reduced token counts. In the following, we analyze various aspects of the results in further detail.

Note that in Table 2, only for `Llama3.2-3B`, Token Distillation obtains results that are on par with random initialization. `Llama3.2-3B(-i)` are the only models in our lineup with tied embedding weights. Our objective does not explicitly enforce a bound on the norm of the new embedding, which in this case led to the failure mode of always generating a specific new embedding with very large norm. Note that this does not always happen for checkpoints with tied weights, as evidenced by the results for `Llama3.2-3B-i`. However, we additionally propose a simple modification to alleviate the occurrence of this issue by combining Token Distillation with the next-token prediction objective, which (implicitly) acts as a regularizer on the output embedding norm. In support of our argument that a subtoken attention distillation objective is superior to next-token prediction for learning new token embeddings, the combination via a sum of the two objectives (Token Distillation + NTP) generally yields worse results than Token Distillation alone. However, we can add a dynamic downweighting factor (see Section 4.2) to the next-token prediction objective (Token Distillation + $\alpha$NTP), which mostly alleviates the negative interference while keeping the regularizing effect.

**Freezing original embeddings.** Vanilla NTP-tuning of all embeddings (denoted as *tune all embeddings*) yields disappointing results, even underperforming the subtoken mean initialization. We investigate this and observe a degradation of the original token embeddings. Note that for Token Distillation, we only optimize new token embeddings. Therefore, we also compare against the NTP baseline, where we similarly optimize only the new token embeddings. NTP outperforms NTP (*tune all embeddings*), supporting our analysis. However, even with this improvement, Token Distillation in turn still further surpasses NTP.

We note that masking gradient updates to the original embeddings during further training is not commonly done when adding new token embeddings, even during initial "embedding tuning" phases where all weights but the embedding layers are frozen. In these phases, the goal is to improve the initialized embeddings for new tokens or "adapt" the new embedding matrix to the existing Transformer layers (de Vries & Nissim, 2021). When normally sampling from a large-scale training corpus, the degradation of original embeddings will be less dramatic, as the number of new tokens per batch will be much lower. However, this is much less computationally efficient and – at least initially – still potentially harmful. We leave a further investigation of this as a practically useful direction for future work.

**Hyper-networks.** ZeTT, which uses a pretrained hyper-network, is the strongest baseline we compare against. We only compare against ZeTT when a pretrained hyper-network by Minixhofer et al. (2024) is available[4]. We first note that ZeTT yields quite impressive results, outperforming even our optimized NTP baseline. Evidently, the pretraining of ZeTT's hyper-network has internalized a better embedding prediction than the iterative gradient descent optimization on selected samples we employ for NTP. Nevertheless, our proposed method Token Distillation outperforms even ZeTT – without needing any hyper-network pretraining. It is important to note that at *inference time*, hyper-network based methods such as ZeTT are actually faster than Token Distillation (and NTP-based baselines), since only a single forward pass through the hyper-network per token is required. However, they require expensive pretraining for each new model and – in some cases – target domain. For example, ZeTT with a hyper-network pretrained only on English naturally underperforms on language adaptation to other languages – as this is out of distribution from the hyper-network training, whereas the multilingually pretrained variant available for one of the models performs more favorably (see Table 3). Also, we can actually use ZeTT-generated embeddings as a starting point and further tune them using our method for even better results. We analyze this possibility in Section 5.3.

## 5.2 COMPRESSION-ORIENTED TOKENS AND CONTINUED TRAINING

In our main experiments, we select new tokens based on frequently occurring *whole words* (yielding moderate sequence compression of 7–10%). In practice, however, vocabulary adaptation is often pursued for computational efficiency, which calls for selecting tokens that maximize sequence compression – even if these tokens are not whole words and are therefore inherently more challenging to initialize well (see our discussion in Appendix A). We therefore include two experiments that explicitly target this regime.

**Compression-oriented tokens on French (Table 4).** We repeat French vocabulary adaptation, but (i) increase the number of snippets per new token from 25 to 100 (4×), and (ii) select 5,000 new tokens using AdaptiVocab (Nakash et al., 2025). Importantly, AdaptiVocab does *not* restrict tokens to whole words, which increases sequence compression but also makes initialization harder (tokens are more ambiguous and more context-dependent than whole words). As a result, *init-only* scores in this experiment are not directly comparable to our main FrenchBench results in Table 3, which use a different (whole-word) token set. AdaptiVocab yields substantially stronger compression, reducing average token counts by about 20% on the evaluated FrenchBench datasets. Despite the increased difficulty of the token set, Token Distillation closely matches the original tokenization while providing the compression benefit.

| Method | Llama-8B-i ($\Delta$ tokens) | Qwen3-8B ($\Delta$ tokens) |
|---|---|---|
| Original tokenization | 72.1 (0%) | **81.7** (0%) |
| Subtoken Mean | 60.9 (**-21%**) | 69.2 (**-20%**) |
| NTP | 66.5 (**-21%**) | 79.1 (**-20%**) |
| ⋆ Token Distillation | **72.2 (-21%)** | 81.5 (**-20%**) |

Table 4: Performance on FrenchBench multiple-choice tasks after adding 5,000 new French tokens selected via AdaptiVocab (Nakash et al., 2025). Token Distillation and NTP use 100 snippets per new token.

| Method | Init only | + Continued training | $\Delta$ tokens |
|---|---|---|---|
| Original tokenization | **47.4** | 46.9 | 0% |
| Subtoken Mean | 36.2 | 45.4 | **-36%** |
| NTP | 40.6 | 46.1 | **-36%** |
| ⋆ Token Distillation | 44.2 | **47.5** | **-36%** |

Table 5: Performance on Arabic multiple-choice tasks after adding 1,000 new Arabic tokens to Mistral-7B selected via AdaptiVocab (Nakash et al., 2025). Token Distillation and NTP use 100 snippets per new token.

**Compression-oriented tokens + continued training on Arabic (Table 5).** We next consider a setting where added tokens come from a language that is very poorly supported in the source tokenizer. We add the top 1,000 Arabic tokens selected via AdaptiVocab to Mistral-7B, reducing token counts on Arabic benchmarks by 36%. Note that the reduction of token counts by 36% corresponds to compute savings of up to 50%, due to the quadratic scaling of attention with respect to sequence lengths (see Appendix B.4). Using the same increased initialization budget as in the previous experiment on French compression-oriented tokens (100 snippets per new token), Token Distillation already outperforms other initialization baselines ("Init only") but does not yet match the original tokenization. We then run a moderate continued-training phase for vocabulary adaptation based on the methodology proposed by Nakash et al. (2025), which trains the embedding matrices as well as the first and last

---

[4]Minixhofer et al. (2024) do not provide a checkpoint trained specifically for the "`-Instruct`" variant of `Llama3-8B` but show that a transfer from the base version is possible.

| | Mistral-7B | Llama3-8B | Llama3-8B-i | Llama3.1-8B | Llama3.1-8B-i | Llama3.2-3B | Llama3.2-3B-i | OLMo2-7B-i | Avg. |
|---|---|---|---|---|---|---|---|---|---|
| Original tokenization | 93.4 | 88.1 | 99.2 | 91.1 | 97.8 | 65.2 | 90.9 | 94.0 | 90.0 |
| Random (Hewitt, 2021) | 0.0 | 0.0 | 0.0 | 0.0 | 0.0 | 0.0 | 0.2 | 0.0 | 0.0 |
| Mean (Gee et al., 2022) | 2.2 | 4.2 | 9.1 | 8.0 | 11.3 | 4.2 | 7.4 | 7.4 | 6.7 |
| NTP (Lampinen et al., 2018) | 34.0 | 54.3 | 70.6 | 49.7 | 71.6 | 29.0 | 53.9 | 70.0 | 54.1 |
| ZeTT (Minixhofer et al., 2024) | 15.3 | 25.4 | 28.4 | – | – | – | – | – | – |
| ⋆Token Distillation | 46.3 | 68.0 | **86.3** | 61.8 | **81.3** | 31.6 | 57.5 | **84.3** | 64.6 |
| ⋆Token Distillation + NTP | **48.3** | 68.0 | 85.3 | 60.2 | 79.7 | **38.2** | 63.4 | 80.9 | 65.5 |
| ⋆Token Distillation + αNTP | 46.9 | **71.0** | 85.7 | **62.0** | 80.1 | 37.2 | **66.4** | 83.7 | **66.6** |

Table 6: Correctness of generated definitions for newly initialized multi-word tokens (famous people, places, entities, sayings and concepts) as judged by `Llama3.3-70B-Instruct`.

| | Mistral-7B | Llama3-8B | Llama3-8B-i | Llama3.1-8B | Llama3.1-8B-i | Llama3.2-3B | Llama3.2-3B-i | OLMo2-7B-i | Avg. |
|---|---|---|---|---|---|---|---|---|---|
| ⋆Token Distillation (generated data) | 62.4 | 67.1 | **67.6** | 67.0 | 70.1 | **57.1** | 62.3 | 60.9 | 64.3 |
| ⋆Token Distillation (retrieved data) | **62.8±0.5** | **67.3±0.2** | **67.6±0.3** | **67.3±0.5** | **71.0±0.2** | 56.2±1.9 | **63.1±0.2** | **61.2±0.3** | **64.6** |

Table 7: Comparison of using data generated from the model instead of retrieving from a corpus (`ncbi/pubmed`). We only run a single seed with generated data and report average performance on biomedical benchmarks.

Transformer layers. After this continued training, Token Distillation matches or slightly exceeds the original tokenization while retaining the substantial token-count reduction. The original tokenization is not able to benefit from this moderate continued training, likely due to heavy over-fragmentation.

## 5.3 In-Depth Analysis

**Multi-word tokens.** In Table 6, we report LLM-as-a-Judge results on the correctness of generated definitions. This tests the limits of our method, as the new tokens are now more complicated, such as "Software Development" or even common phrases such as "spill the beans". In this setting, even using the original tokenization sometimes is not able to generate correct definitions. In turn, the gap of Token Distillation to the original tokenization is also larger in this experiment. However, Token Distillation outperforms all other baselines by a very large margin. ZeTT (Minixhofer et al., 2024) in particular struggles in the multi-word token setup, likely because it is out-of-distribution from the hyper-network pretraining.

**Retrieved vs. generated data.** In Section 3, we discuss generating contexts containing our new tokens by prompting the model to generate some text containing the new token. In our main experiments, we instead retrieve relevant contexts from a corpus. We compare the two approaches in Table 7. In general, data generation performs slightly worse than retrieving contexts but is still competitive, outperforming next-token prediction objective on retrieved contexts and even pretrained hyper-works from ZeTT. This highlights the applicability of our method even in scenarios where no corpus containing relevant contexts might be available. For example, we use context generation instead of retrieval in our multi-word token experiments in Table 6.

**Choice of initial embedding for Token Distillation.** We find that the subtoken mean initialization as starting point yields better results than random initialization (see "Random + Token Distillation" in Table 8). We use the subtoken mean for its simplicity. In settings where new tokens have low lexical overlap with existing tokens, better results could be achieved via mapping-based initialization methods (*e.g.*, Minixhofer et al., 2022; Dobler & de Melo, 2023). If a pretrained embedding prediction hyper-network is available, we can instead use these predicted embeddings as the starting point. We run this experiment using ZeTT; this combination further improves performance compared to using the subtoken mean (see Table 8). We also compare refining the hyper-network embeddings with NTP. This greatly improves over NTP but yields only minimal gains over using the hyper-network embeddings themselves without further NTP.

**Different distillation objectives.** Our main proposed method Token Distillation computes the distillation objective based on a mean-squared error (MSE) of last layer hidden states. A natural question is whether we can also use the "traditional" knowledge distillation objective (Hinton et al., 2015) based on the Kullback-Leibler divergence (KL) (Kullback & Leibler, 1951). We analyze this and report the results in Table 9. We additionally include computing the MSE between logits instead of hidden states. For the KL-based results, we include different variations of combining the objective with a NTP-based objective. In general, the choice of distillation objective does not

matter much. This is hardly surprising, as the logits are merely a linear projection (potentially with an additional normalization layer) of the hidden state; the log-probabilities used for KL are then in turn merely an additional log-normalization via log-softmax. Combining the KL-based results with NTP-based objectives follows similar trends as their combinations with hidden state MSE-based objectives. We opt for the MSE on hidden states for two reasons: (1) In terms of implementation convenience, the objectives with a projection to logits over the vocabulary require a masking step for the "student" model, as the original "teacher" model does not have the new tokens in its vocabulary. (2) Choosing hidden states allows us to explore other layers than just the last one with potentially large computational speedups – we explore this next.

| | Mistral-7B | Llama3-8B | Llama3-8B-i | Avg. |
|---|---|---|---|---|
| Original Tokenization | 64.5 | 69.8 | 70.6 | 68.3 |
| Random (Hewitt, 2021) | 56.7 | 58.7 | 60.6 | 58.6 |
| Mean (Gee et al., 2022) | 58.3 | 63.8 | 63.4 | 61.8 |
| ZeTT (Minixhofer et al., 2024) | 62.7 | 66.1 | 66.3 | 65.0 |
| (Mean +) NTP | 61.5 | 65.9 | 65.4 | 64.3 |
| ZeTT + NTP | 62.8 | 66.1 | 66.8 | 65.3 |
| ⋆ Random + Token Distillation | 62.5 | 65.3 | 65.9 | 64.6 |
| ⋆ (Mean +) Token Distillation | 63.5 | 67.3 | 68.0 | 66.3 |
| ⋆ ZeTT + Token Distillation | **64.2** | **67.9** | 68.4 | 66.8 |
| ⋆ (Mean +) Token Distillation + $\alpha$NTP | 63.2 | 67.2 | 68.4 | 66.3 |
| ⋆ ZeTT + Token Distillation + $\alpha$NTP | 64.1 | **67.9** | **68.7** | **66.9** |

Table 8: Biomedical benchmark results with different starting points for Token Distillation. We report results for models where a pretrained hyper-network from ZeTT is available and consider the same single seed for all methods.

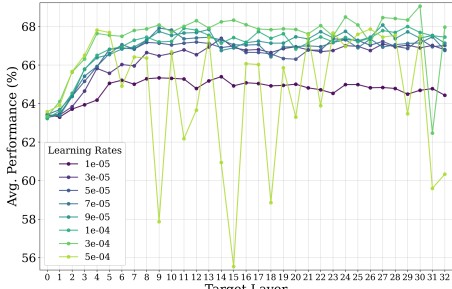

Figure 2: Average performance on biomedical benchmarks with different target layers of Llama3-8B-Instruct for Token Distillation.

**Target layer.** In our main experiments, we apply the Token Distillation objective on the hidden states after the last layer. However, the last hidden state might not be the most optimal, as it might be overspecialized for next-token prediction (Rogers et al., 2020). In Figure 2, we plot the average performance on our biomedical benchmark datasets using `Llama3-8B-Instruct` while varying the target layer for extracting hidden states for our Token Distillation objective. Indeed we do see a slight downward trend as we approach the last layer – choosing a different layer than the last one can further boost our already strong results. We repeat the analysis with `Llama3.1-8B-Instruct` in Appendix B.3 with matching results.

Interestingly, good results can already be achieved with very early layers, *e.g.*, after layer four. This suggests that much of the subtoken-specific contextual information is already added to the residual stream in early layers, which echoes findings of previous work (Vulić et al., 2020; Elhage et al., 2022; Lad et al., 2024; Nakash et al., 2025; Kaplan et al., 2025). This opens up an avenue to further speed up our method by terminating the forward pass after our target layer (and also saving that compute on the backward pass), which can be much faster if we select early target layers. We leave further exploration of this as an exciting direction for future work. For our main experiments, we choose to keep the last layer because this choice does not necessitate a model-specific sweep over target layers. Also, the last layer is a principled choice, as it guarantees that no subtoken interactions that are only modeled in later layers are excluded from the objective.

## 6  CONCLUSION

In this work, we have described a fundamental limitation of many existing embedding initialization methods, which only exploit knowledge stored in a model's embedding matrices, whereas much of the knowledge about token compositions actually resides in the Transformer layers (Elhage et al., 2022; Lad et al., 2024). To address this limitation, we have proposed Token Distillation, which systematically incorporates this knowledge via distilling contextual information from hidden states. Experimental results confirm that our method not only clearly outperforms methods aggregating embedding matrix rows but also yields better performance than training embeddings using traditional causal language modeling as well as hyper-networks extensively pretrained for new token embedding prediction (Minixhofer et al., 2024). In future work, we believe research into embedding initialization should move beyond an aggregation of information stored in the embedding tables. In particular, a more localized identification of subtoken contextualization (*e.g.*, specific attention heads that aggregate subtokens) is a promising direction for a more targeted distillation objective.

ACKNOWLEDGEMENTS

We thank the German Federal Ministry of Research, Technology and Space for their compute grant through the project «KI-Servicezentrum Berlin Brandenburg» (16IS22092). Konstantin Dobler further thanks the European Laboratory for Learning and Intelligent Systems (ELLIS) PhD program for support. The research was supported by a research grant (VIL53122) from VILLUM FONDEN, and by the European Union's Horizon 2020 research and innovation program under grant agreement No. 101135671 (TrustLLM).

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

## A  LIMITATIONS

We have already highlighted and analyzed various choices and potential weaknesses of our method in Section 5.1 and Section 5.3. Nevertheless, we find that our method outperforms even very strong baselines. However, we now want to explicitly summarize and expand on the limitations of our method.

**Unknown tokens.** Since the motivation for the distillation-based objective is to match the original model's behavior, our method is generally not applicable if the original model cannot meaningfully process a new token. Note that this specifically limits the applicability of our method for adapting a model's vocabulary to a completely unseen language, although in current pretraining regimes, many languages do end up being seen to a certain extent due to imperfect filtering mechanisms or deliberate inclusion.

**Learning rate.** Since our method involves gradient descent, we require setting a learning rate. However, as can be seen in Figure 2, our method seems to afford a wide range of viable learning rates that achieve good results, rendering the tuning of this hyperparameter less critical.

**Only input embeddings.** As discussed in Section 3, our method only addresses the task of inducing *input embeddings*, since our distillation objective is not applicable to output embeddings. Our method can be freely combined with any other method for obtaining valid output embeddings. Alternatively, new tokens can be added only in the input embeddings, since the Transformer architecture does allow for input tokens which do not occur in the output vocabulary (although this assumption is made in some popular implementations).

**Short subword tokens.** We find that our method does not work as well for lexically ambiguous tokens, *i.e.*, tokens with many different meanings in different contexts. These are often short subwords like `<ed>` or `<fer>` that occur in many words. This also means that our method is not ideal to initialize embeddings for an entirely new vocabulary that contains such new subwords. However, these tokens

could be initialized using other methods (*e.g.* Minixhofer et al., 2024), while our method is applied to the rest.

**Impact of chosen reference snippets.** The reference corpus for retrieving contexts for training with the distillation objective can bias the resulting learned embeddings. Consider the word "tender", which in general means gentle or soft, but in the financial domain describes a specific type of buy offer. Note however that this can also be beneficial, as the resulting representations are now specialized to the target domain. Additionally, if such an effect is undesired, we can use our proposed data generation scheme via prompting the model to generate contexts containing the new word. This can be interpreted as distillation training on samples from the original model's learned data distribution of samples containing the new word.

## B  ADDITIONAL RESULTS

### B.1  DIFFERENT DISTILLATION OBJECTIVES

We report results for using different distillation objective variants in Table 9. The difference between hidden state or logit-based distillation as well as the differences between MSE and KL based distillation is not significant. However, our chosen approach of hidden state distillation using MSE has the additional advantage of allowing the use of earlier hidden states, offering significant speedups (see Figure 2 and Figure 3).

| | Mistral-7B | Llama3-8B | Llama3-8B-i | Llama3.1-8B | Llama3.1-8B-i | Llama3.2-3B | Llama3.2-3B-i | OLMo2-7B-i | Avg. |
|---|---|---|---|---|---|---|---|---|---|
| ⋆Token Distillation (Logits) | **63.0±0.4** | 67.2±0.3 | 67.5±0.8 | 67.3±0.2 | 70.7±0.5 | 57.4±0.3 | 62.8±0.7 | 60.2±0.2 | 64.5 |
| ⋆Token Distillation + NTP | 63.0±0.5 | 67.2±0.3 | 66.7±1.6 | 67.2±0.3 | 70.6±0.4 | 57.6±0.3 | 62.2±0.3 | 59.8±0.5 | 64.3 |
| ⋆Token Distillation + αNTP | 62.8±0.5 | **67.6±0.4** | **67.8±0.5** | **67.4±0.4** | 70.9±0.3 | **57.9±0.1** | 62.5±0.6 | **61.2±0.2** | **64.7** |
| ⋆Token Distillation | 62.8±0.5 | 67.3±0.2 | 67.6±0.3 | 67.3±0.5 | **71.0±0.2** | 56.2±1.9 | **63.1±0.2** | 61.2±0.3 | 64.6 |
| ⋆Token Distillation (KL) + NTP | 62.3±0.4 | 66.7±0.2 | 66.4±0.3 | 66.7±0.7 | 70.1±0.4 | 57.3±0.2 | 62.1±0.6 | 59.7±0.3 | 63.9 |
| ⋆Token Distillation (KL) + αNTP | 62.8±0.5 | 67.1±0.4 | 66.3±1.2 | 67.1±0.2 | 70.5±0.4 | 57.5±0.5 | 62.5±0.2 | 60.5±0.5 | 64.3 |
| ⋆Token Distillation (KL) | 63.0±0.2 | 67.3±0.2 | 67.2±0.2 | 67.1±0.2 | 70.8±0.5 | 57.5±0.4 | 62.7±0.3 | 60.5±0.6 | 64.5 |

Table 9: Comparison of different variations of our distillation objective with results on biomedical benchmarks. We **bold** the best result in each column and underline all results that are not significantly worse (one-sided Welch's t-test with Bonferroni correction, $p < 0.05$). ⋆: Our method(s).

### B.2  RESULTS FOR "TOKENS TO WORDS" (KAPLAN ET AL., 2025)

We report results for the method from "Tokens to Words" (Kaplan et al., 2025) in Table 10 and Table 11 alongside our method and other baselines. Note that the "Tokens to Words" method only initializes embeddings for tokens where a PatchScopes (Ghandeharioun et al., 2024) extraction using a special prompt to identify hidden layers is successful. It is difficult to apply their method if the extraction is not successful. To give the fairest assessment of the method, we do not add these "unsuccessful" tokens to the vocabulary when using "Tokens to Words". However, this likely

| | Mistral-7B | Llama3-8B | Llama3-8B-i | Llama3.1-8B | Llama3.1-8B-i | Llama3.2-3B | Llama3.2-3B-i | OLMo2-7B-i | Avg. |
|---|---|---|---|---|---|---|---|---|---|
| Original tokenization | 64.5 | 69.8 | 70.6 | 69.2 | 72.3 | 60.1 | 64.4 | 61.3 | 66.5 |
| Random (Hewitt, 2021) | 57.0±0.6 | 58.8±0.5 | 60.6±0.3 | 59.3±0.5 | 62.6±0.6 | 51.6±0.4 | 55.8±0.2 | 53.9±0.6 | 57.5 |
| NTP (*tune all embeddings*) | 55.7±0.5 | 63.8±0.1 | 63.8±0.2 | 62.9±0.4 | 66.8±0.4 | 51.5±0.6 | 55.5±0.5 | 58.0±0.3 | 59.8 |
| Mean (Gee et al., 2022) | 58.3 | 63.8 | 63.4 | 63.7 | 66.9 | 54.2 | 58.4 | 58.0 | 60.8 |
| Tokens to Words (Kaplan et al., 2025) | (61.7) | (64.3) | (65.0) | (63.9) | (66.6) | (57.3) | (61.5) | (59.5) | (62.5) |
| NTP (Lampinen et al., 2018) | 61.2±0.4 | 65.6±0.3 | 65.3±0.6 | 66.0±0.5 | 68.9±0.2 | 56.6±0.6 | 61.3±0.5 | 58.7±0.6 | 63.0 |
| ZeTT (Minixhofer et al., 2024) | 62.7 | 66.1 | 66.3 | – | – | – | – | – | – |
| ⋆Token Distillation + NTP | **63.0±0.5** | 67.2±0.3 | 66.7±1.6 | 67.2±0.3 | 70.6±0.4 | 57.6±0.3 | 62.2±0.3 | 59.8±0.5 | 64.3 |
| ⋆Token Distillation | 62.8±0.5 | 67.3±0.2 | 67.6±0.3 | 67.3±0.5 | **71.0±0.2** | 56.2±1.9 | **63.1±0.2** | 61.2±0.3 | 64.6 |
| ⋆Token Distillation + αNTP | 62.8±0.5 | **67.6±0.4** | **67.8±0.5** | **67.4±0.4** | 70.9±0.3 | **57.9±0.1** | 62.5±0.6 | **61.2±0.2** | **64.7** |

Table 10: Benchmark results on biomedical domain adaptation for different initialization methods. We report a macro-average ± standard deviation over five random seeds on the tasks in the Open Medical-LLM leaderboard (see Section 4.1). The best initialization result for each model is given in **boldface**, while all results that are not significantly worse (one-sided Welch's $t$-test with Bonferroni correction, $p < 0.05$) are underlined. Methods without ±x.x are deterministic. −: We only report results for ZeTT where pretrained hyper-networks are available. (...): Tokens to Words is able to extract representations for only 60%–90% of added biomedical tokens. We still report results for completeness but note that this comparison overestimates the method's performance compared to the other methods shown. ⋆: Our method(s).

|  | Mistral-7B | Llama3-8B | Llama3-8B-i | Qwen3-8B | Avg. |
|---|---|---|---|---|---|
| Original tokenization | 69.5 | 69.4 | 72.1 | 81.7 | 73.2 |
| Random (Hewitt, 2021) | 51.0 | 46.1 | 51.5 | 62.1 | 52.7 |
| Mean (Gee et al., 2022) | 56.3 | 58.4 | 61.7 | 69.6 | 61.5 |
| Tokens to Words (Kaplan et al., 2025) | (64.8) | (62.6) | (66.6) | (**82.2**) | (69.1) |
| NTP (Lampinen et al., 2018) | 64.7 | 67.0 | 70.1 | 81.2 | 70.8 |
| ZeTT (Minixhofer et al., 2024) | 66.1 | 61.3[†] | 65.2[†] | – | – |
| ⋆ Token Distillation | **68.5** | **68.9** | **72.9** | 81.5 | **72.9** |

Table 11: Performance on FrenchBench multiple-choice tasks. [†]: For ZeTT, only Mistral-7B has a multilingual-tuned hyper-network, while Llama3-8B(-i) uses the general English+Code hyper-network. (...): Tokens to Words is able to extract representations for only 25%–50% of added French tokens. We still report results for completeness but note that this comparison overestimates the method's performance compared to the other methods shown.

overestimates the method's performance, since the model now uses more of the original embeddings during testing, which have been tuned during massive pretraining. On average on our biomedical domain adaptation results, roughly 10–20% of the total tokens did not have a successful PatchScopes extraction, varying across models (up to 35% in the case of Llama3.2-3B and 40% in the case of OLMo2-7B-i). On our French language adaptation experiments, this is even more severe, ranging from 50% to 75% of unsuccessful extractions.

### B.3 DIFFERENT TARGET LAYERS FOR TOKEN DISTILLATION

We repeat the analysis of the best target layer for Token Distillation from Section 5.3 in Figure 3 for `Llama3.1-8B-Instruct` instead of `Llama3-8B-Instruct`. Our analysis for `Llama3-8B-Instruct` in Section 5.3 applies.

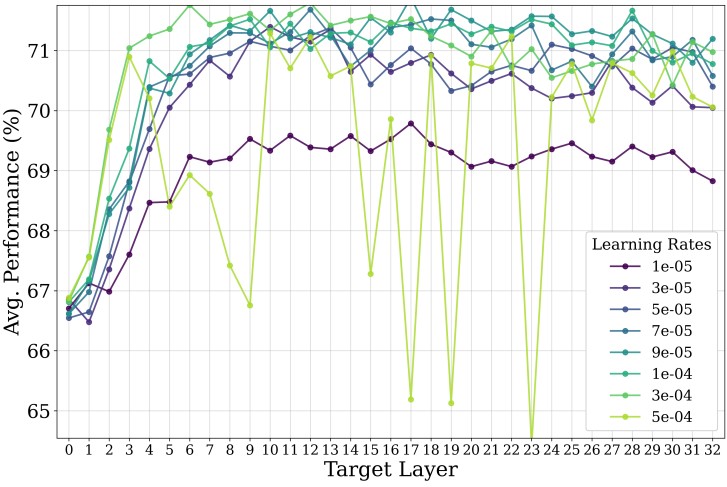

Figure 3: Analysis of varying the target layer for the Token Distillation objective. We report the average performance on the biomedical benchmarks for `Llama3.1-8B-Instruct`.

### B.4 WALL-CLOCK SPEEDUPS FROM REDUCED OVER-TOKENIZATION

**A note on downstream computational speedups.** Over-tokenization is not only associated with reduced downstream task performance (Rust et al., 2021; Ali et al., 2024) but also with increased computational cost due to longer sequence lengths (Ahia et al., 2023; Yamaguchi et al., 2024a). This computational cost is reduced when reducing over-tokenization by adding new (commonly occurring) tokens to the vocabulary. Importantly, for input embeddings, the reduction in computational cost is completely independent of the actual quality of the embeddings for the newly added tokens (for a given fixed sequence of text). The same computational speedups are achieved by using Random initialization and much better methods such as ZeTT (Minixhofer et al., 2024) or our proposed Token Distillation. Of course, the actual quality of any predictions made conditioned on a sequence

containing such new tokens does heavily depend on the embedding quality. Therefore, in this work, we primarily focus on direct evaluation of the quality of produced embeddings. However, we further provide an empirical wall-clock validation of the efficiency gains next.

**Empirical speedups.**    We empirically validate that reduced token counts translate into concrete wall-clock speedups. We compare batch processing of documents from the `arb_Arab` split of `HuggingFaceFW/fineweb-2` between (i) the original Mistral-7B model and tokenizer and (ii) an adapted model using Token Distillation with 1,000 Arabic tokens selected via AdaptiVocab (Nakash et al., 2025). We repeat the benchmark on both Nvidia H100 GPUs and Nvidia B200 GPUs. We use a single GPU, process 8192 documents per run, and truncate documents to 4k characters. For B200 GPUs, we use a batch size of 4 and for H100 GPUs, we use a batch size of 1 to avoid OOM. We report both inference-style forward-only runtimes and training-style forward+backward runtimes. Note how the compute savings can exceed the token-count reduction due to the quadratic attention cost in sequence length.

| Accelerator | $\Delta$ tokens | $\Delta$ compute (fwd) | $\Delta$ compute (fwd+bwd) | $\Delta$ tok/sec (fwd) | $\Delta$ tok/sec (fwd+bwd) |
|---|---|---|---|---|---|
| H100 | -35% | -41.7% | -40.8% | +9.5% | +7.9% |
| B200 | -35% | -49.7% | -50.5% | +24.6% | +26.2% |

Table 12: Measured throughput and wall-clock compute savings from reduced over-tokenization on Arabic documents.

## C    IMPLEMENTATION DETAILS

### C.1    TOKEN SELECTION FOR DOMAIN ADAPTATION

When using benchmarks to evaluate the quality of new token embeddings, it is crucial to ensure that the new tokens actually occur frequently enough to affect the performance if the new embeddings are poor. See, *e.g.*, the "Random" baseline for randomly initialized embeddings, which still performs better than random chance on the benchmarks in Table 1.

Therefore, we select all whole words from the chosen benchmarks that are not already existing tokens in the original vocabulary. We exclude tokens that include digits or any of the characters in the following string in order to reduce noise:

```
"[]{}()<>.,;:!?@#$%^&*+_=|\/"
```

Additionally, to exclude very rare words that would not be commonly used as new tokens, we apply the following two filters: (1) We exclude words that appear fewer than five times across all benchmarks, and (2) we exclude words that occur fewer than 25 times in a sample of 768,000 documents from a domain-specific reference corpus (we use `ncbi/pubmed` for biomedical benchmarks). This amounts to roughly 2,600 new tokens (varying per model). The full list of new words is available in our GitHub repository. For illustrative purposes, we provide a random sample of 100 added tokens here:

```
[gangrene, emphysema, Alkaline, pleural, psychoanalysis, paracervical,
↪  glandular, acidosis, reticular, two-factor, hyperplasia, bicarbonate,
↪  atopy, reactant, Riboflavin, lipoprotein, long-term, Chediak-Higashi,
↪  attenuated, pruritus, immunohistochemistry, calcifications,
↪  Staphylococcal, aspirin, bromide, mechanistic, cytosol, insufficiency,
↪  arouse, thyroidectomy, Fallopian, primum, retrograde, febrile,
↪  Cytomegalovirus, molar, cryptogenic, inorganic, deceleration,
↪  Amikacin, gluconate, inhaler, Cushing, discordant, Epstein-Barr,
↪  intraocular, VLDL, tomography, supraclavicular, glucocorticoids,
↪  pyruvate, occluded, pigmented, neutrophil, irradiation,
↪  pharmaceuticals, cementum, LDL-cholesterol, postpartum, contractile,
↪  prostatectomy, deafness, septic, analgesics, tonsils, side-to-side,
↪  Prostaglandin, hoarseness, acuity, vena, dislocated, inciting,
↪  haloperidol, eczema, remnant, innervation, heartburn, elicited,
↪  yellowish, amylase, antihypertensive, ossification, redness, Giardia,
↪  placenta, trigeminal, Spearman, squamous, perioperative, abdominis,
↪  photosynthetic, transcribed, thymoma, condylar, furosemide, exudate,
↪  kJ, afebrile, non-exposed, unrestrained]
```

## C.2 TOKEN SELECTION FOR FRENCH LANGUAGE ADAPTATION (TABLE 3)

We use a similar approach as described in Appendix C.1 but use `HuggingFaceFW/fineweb-2` as a domain-specific reference corpus for filtering rare words and add words that appear in our chosen benchmarks from FrenchBench (Faysse et al., 2025). This amounts to roughly 11,600 new tokens (varying per model). We also provide a random sample of 100 added tokens for illustrative purposes here:

```
[excitation, pelvienne, biblique, préserver, ratisse, coiffants, narine,
↪ mutuellement, sentira, ventilateurs, Réalisez, Accédez, relaient,
↪ inclinée, trébuche, structurer, pissenlit, contenants, débutant,
↪ annulation, mucus, accueillir, grossiers, sérums, œuf, Enfin,
↪ victoire, recevrez, balayer, vendez, officiel, agression, Penser,
↪ suspendue, versez, socialement, scientifiques, chirurgical,
↪ Voulez-vous, doués, poursuivi, glacière, boîtier, purifier, rocheuse,
↪ profitant, connaissent, partenaires, cuisent, paresseux,
↪ considérations, mèche, substitut, relaxante, célébrité, culpabilité,
↪ visuel, convulsions, correcteurs, en-dessous, alcoolisée,
↪ développeront, mètres, fabriquer, imperméables, vapeur, granuleux,
↪ lier, nuage, creusez, opérations, entourent, ressentant, extérieurs,
↪ intenter, métropolitaine, chutes, arrière-plan, miettes, flaque,
↪ renforcera, entonnoir, acheté, brillant, courtier, kiwi, texturée,
↪ brouette, diminuera, débutants, plaire, quitté, appétit, abîmer,
↪ équilibrée, avancez, pratiquent, inconvénient, restants, offriront]
```

## C.3 MULTI-WORD TOKEN GENERATION

We prompted `gpt-o4-mini-high` (as of April 7, 2025) with the following prompt:

```
f"""Provide a list of entities, places, people, concepts, phrases or
↪ phrasings, idioms, or other terms that span multiple words, at least
↪ two and maximum five. Provide them in JSON format grouped by category
↪ and at least 100 per category. Ensure that you do not shorten
↪ coherent names just to fit them into the five word limit."""
```

We refined this prompt to exclude common failure modes through trial-and-error. We include the list of resulting multi-word tokens in our GitHub repository.

## C.4 OPEN MEDICAL-LLM LEADERBOARD EXPERIMENTS

For evaluation, we use the `lm-evaluation-harness` (Biderman et al., 2024) library at the version `0.4.7`. We use 5-shot evaluation on a single Nvidia H100 GPU using `bfloat16` precision. Specifically, we use the following command:

```
lm_eval --model hf \
        --model_args pretrained=$MODEL_PATH,dtype="bfloat16" \
        --tasks
        ↪ medmcqa,medqa_4options,mmlu_anatomy,mmlu_clinical_knowledge,
        ↪ mmlu_college_biology,mmlu_college_medicine,mmlu_medical_genetics,
        ↪ mmlu_professional_medicine,pubmedqa \
        --num_fewshot 5 \
        --device cuda:0 \
        --batch_size 8
```

This evaluates the task of the Open Medical-LLM Leaderboard (Pal et al., 2024; Jin et al., 2019; 2020; Hendrycks et al., 2021; Pal et al., 2022). All code and a lock-file with specific versions are available in our GitHub repository.

## C.5 FRENCHBENCH EXPERIMENTS

As for the Open Medical-LLM Leaderboard Experiments, we use the `lm-evaluation-harness` (Biderman et al., 2024) library at the version `0.4.7` and use 5-shot evaluation on a single Nvidia H100 GPU with `bfloat16` precision. We evaluate on the multiple-choice tasks from FrenchBench (Faysse et al., 2025) and use letters (*e.g.*, A, B, C or D) as the answer choices. We publish the task configuration in our GitHub repository. Specifically, we use the following command:

```
lm_eval --model hf \
        --model_args pretrained=$MODEL_PATH,dtype="bfloat16" \
        --include_path ./paper/evals/frenchbench_abcd/ \
        --tasks french_bench_boolqa_ab,french_bench_vocab_abcd,
    ↪   french_bench_xnli_abc,french_bench_arc_challenge_abcd,
    ↪   french_bench_grammar_abcd,french_bench_fquadv2_bool_ab,
    ↪   french_bench_topic_based_nli_abc,
    ↪   french_bench_reading_comp_abcd \
        --num_fewshot 5 \
        --device cuda:0 \
        --batch_size 8
```

## C.6 ARABIC EXPERIMENTS

We use the `lm-evaluation-harness` (Biderman et al., 2024) library at the version `0.4.7` and use 5-shot evaluation on a single Nvidia H100 GPU with `bfloat16` precision. We evaluate on multiple-choice tasks from the `arabic_leaderboard_light` (Elfilali et al., 2024) and use letters (*e.g.*, A, B, C or D) as the answer choices. We publish the task configuration in our GitHub repository. Specifically, we use the following command:

```
lm_eval --model hf \
        --model_args pretrained=$MODEL_PATH,dtype="bfloat16" \
        --include_path ./paper/evals/arabic_leaderboard_light_abcd \
        --tasks arabic_leaderboard_light_abcd \
        --num_fewshot 5 \
        --device cuda:0 \
        --batch_size 8
```

## C.7 LLM-AS-A-JUDGE EXPERIMENTS

For generating definitions for a particular new token, we use the prompt "The word `<new_token>` is defined as" following Teehan et al. (2024), where `<new_token>` is replaced by the string representation of the new token (with whitespace stripped from the left side). We use greedy decoding. Our evaluation considers the model checkpoint corresponding to the best learning rate from the experiments described in Appendix C.4 for each method.

LLM-as-a-Judge evaluations are not perfect – see for example in Table 2 the similarity scores (Sim) of the target model (measuring similarity *with the original tokenization*), which should obviously always be 100%, but turn out to be 99.4% / 99.8% in the case of `OLMo2-7B-i` and `Mistral-7B`, respectively. Nevertheless, these results (which we report as a sanity check on the judge quality) are very close to correct, which is encouraging.

We use `meta-llama/Llama3.3-70B-Instruct` as the judge model and run it using `bfloat16` layer-parallel inference over two H100 GPUs using HuggingFace `transformers`. We use prompts adapted from Li et al. (2023) and Villegas et al. (2025) for evaluating the general correctness of generated definitions as well as the similarity with the original model.

For correctness evaluation, we use the following prompt:

```
f"""<|start_header_id|>system<|end_header_id|>

You are a pattern-following assistant that can only answer with "Yes" or
↪   "No". Your goal is to determine whether a provided definition for a
↪   given word is correct. The definition should be on topic and specific
↪   but does not need to be
↪   exhaustive.<|eot_id|><|start_header_id|>user<|end_header_id|>

Remember to answer with one word either "Yes" or "No".

### Instruction:
Determine if the provided definition for the word "{token.lstrip()}" is
↪   correct.

### Definition {token.lstrip()}:
{line["model_definition"].strip()}
```

```
### Is the provided definition correct, specific, and on topic (Yes or
↪ No)?:<|eot_id|><|start_header_id|>assistant<|end_header_id|>\n\n"""
```

For similarity with the original model's definition, we use this prompt:

```
f"""<|start_header_id|>system<|end_header_id|>

You are a pattern-following assistant that can only answer with "Yes" or
↪ "No". Your goal is to determine whether a predicted definition
↪ conveys a similar enough meaning to the ground truth definition
↪ provided for a given
↪ word.<|eot_id|><|start_header_id|>user<|end_header_id|>

Remember to answer with one word either "Yes" or "No".

### Instruction:
Determine if the predicted definition conveys a similar meaning to the
↪ ground truth definition. The word is "{token.lstrip()}".

### Ground truth definition:
{line["target_definition"].strip()}

### Predicted definition:
{line["model_definition"].strip()}

### Does the predicted definition convey a similar meaning to the ground
↪ truth definition (Yes or
↪ No)?:<|eot_id|><|start_header_id|>assistant<|end_header_id|>\n\n"""
```

### C.8 GENERATION OF RELEVANT CONTEXTS

When generating contexts that contain a new token, we prompt the model with the beginning-of-sequence (BOS) token followed by the textual representation of the new token. Using the new token `<palatable>` as an example, our prompt is then: "`` palatable". Note that we add a prefix space after the BOS token ``. We then sample $N$ new sequences of length $L$, where $N$ and $L$ are the same as in the settings we use for context retrieval from an existing corpus (in this work we use $N = 25$ and $L = 50$). We sample using the standard sampling settings of each model from the `transformers` (Wolf et al., 2020) library.

### C.9 ISOLATED PER-TOKEN OPTIMIZATION VS. JOINT OPTIMIZATION

When implementing Token Distillation, we have the choice of performing the optimization process for each new input embedding separately or simply allowing updates to all new input embeddings at the same time.

For the main experiments in this paper, we utilize the joint optimization approach. However, we also implemented the isolated optimization of each new embedding for initial experiments. In this setting, gradients are only propagated to a single input embedding (corresponding to the target token of the retrieved snippet), even if other new target tokens also occur in that snippet.

In our exploratory experiments, we found that the joint approach yields slightly better results and conjecture that this is because we are able to have more total gradient updates per target token due to co-occurrence in a particular snippet. As an added benefit, the joint approach has less implementation complexity. However, when most tokens are actually new tokens (such as in our experiment with adaptation of `Mistral-7B-v0.1` to Arabic in Section 5.2), the separate approach should be preferred because the student model mostly sees "noisy" new embeddings instead of just a single one per sequence.

### C.10 HYPERPARAMETERS

We use AdamW (Kingma & Ba, 2017; Loshchilov & Hutter, 2019) optimization for all trainable parameters (the new token embeddings) with a batch size of 16, which maximizes throughput on our used Nvidia H100 80GB GPUs. We do not incorporate any weight decay and maintain a constant learning rate schedule with a linear warmup for the first half of the training steps. For fair comparison,

we sweep for the best learning rate for each method that requires a learning rate. We use the same data for all training-based methods, including baselines and variations of our method.

## C.11 Continued Training Hyperparameters for Arabic Vocabulary Adaptation

For the continued-training setting in Table 5, we follow a lightweight vocabulary adaptation protocol based on AdaptiVocab (Nakash et al., 2025). We train the embedding matrices as well as the first and last Transformer layers. We use an effective batch size of 128 sequences of max length 2,048 and train for 1,000 steps (about 260M training tokens). We use AdamW with a learning rate of $5 \cdot 10^{-5}$, 100 warmup steps, and cosine decay.

## C.12 Models

We use the models `Mistral-7B-v0.1` (Jiang et al., 2023), `OLMo2-7B-1124-Instruct` (OLMo et al., 2025), `Llama3-8B`, `Llama3-8B-Instruct`, `Llama3.1-8B`, `Llama3.1-8B-Instruct`, `Llama3.2-3B`, `Llama3.2-3B-Instruct` (Grattafiori et al., 2024) and `Qwen3-8B-Base` (Yang et al., 2025).

# D Compute Budget & Runtime

We report the average training time for Token Distillation as well as NTP for initializing the new biomedical domain tokens per model in Table 13. We also report the number of new tokens per model. These differ slightly, as we build an initial fixed candidate list, which is then filtered against the existing vocabularies of each model.

**Additional computational budget.** Since Token Distillation includes an additional forward pass through the model for generating "teacher" hidden states, we do see this overhead reflected in the run times. A forward pass can be approximated as taking one third of the computational cost of a complete forward-backward pass – which roughly matches our measured run times. Note however that – as demonstrated in Section 5.3 – Token Distillation can be significantly sped up while potentially even improving results by choosing an earlier target layer than the last one, yielding even faster run times than NTP.

However, our main version of Token Distillation is slightly more expensive than NTP – even though both have the same data budget. Thus, we also run the baseline NTP with an *additional epoch* to compensate for the additional teacher forward pass that Token Distillation uses. Note that this in fact significantly overcompensates for the additional budget Token Distillation uses. We report the results in Table 14. NTP is *not able* to take any major advantage of an additional epoch, yielding similar results to using a single epoch. Investigating the loss curves, we find that the training converges towards the beginning of the second epoch.[5]

We believe that this is at least partly due to the fundamental limitation of next-token prediction for our task of learning a single new embedding for an already pretrained model. Instead of matching model behavior between seeing the single new and the multiple original subtokens, NTP simply adjusts weights to maximize the likelihood of the given sequences. Consider "`<new_token>` *is a football player*", where we want `<new_token>` to represent `<Messi>`: NTP will only be able to learn a generic embedding from this sequence that will capture semantics from many different (American football *and* soccer) players. Our proposed distillation objective Token Distillation however is able to learn a more specific representation, as it has access to hidden states from the sequence "*Messi is a football player*", which capture more specific details necessary for a good representation.

To demonstrate, we also run Token Distillation for an additional epoch. Token Distillation is in fact able to take advantage of a second pass over the data, in some case with major improvements in the benchmark results. This illustrates the richer training signal that our distillation-based objective provides. Remarkably, for `OLMo2-7B-i`, Token Distillation now is actually able to match the original model with the original tokenization while instead using the new tokenization.

**Token Distillation versus hyper-networks.** Comparing the computational budget with ZeTT (Minixhofer et al., 2024), ZeTT sees 3.2 *billion* tokens during hyper-network pretraining, while Token Distillation sees only 3.2 *million* tokens (for initializing 2,600 new tokens), less than 1% of ZeTT's

---

[5]Note that we do not anneal the learning rate to zero, so this is not an artifact of the learning rate schedule.

| Init Method | Llama3.1-8B | Llama3.1-8B-i | Llama3.2-3B | Llama3.2-3B-i | Llama3-8B | Llama3-8B-i | Mistral-7B | OLMo2-7B-i |
|---|---|---|---|---|---|---|---|---|
| NTP | 08:44 | 08:49 | 05:52 | 05:18 | 08:43 | 08:46 | 06:01 | 07:40 |
| Token Distillation | 11:31 | 11:33 | 07:38 | 08:07 | 12:00 | 11:29 | 09:09 | 10:22 |
| # new tokens | 2589 | 2589 | 2589 | 2589 | 2589 | 2589 | 2637 | 2591 |

Table 13: Minimum training times (mm:ss) on the domain adaptation token initialization experiments for each model and initialization method measured over three runs. We report the minimum, as we run experiments on a shared cluster. We use a single H100 80GB GPU. We also include the number of new tokens for each model type.

| | Mistral-7B | Llama3-8B | Llama3-8B-i | Llama3.1-8B | Llama3.1-8B-i | Llama3.2-3B | Llama3.2-3B-i | OLMo2-7B-i | Avg. |
|---|---|---|---|---|---|---|---|---|---|
| Original tokenization | 64.5 | 69.8 | 70.6 | 69.2 | 72.3 | 60.1 | 64.4 | 61.3 | 66.5 |
| NTP (1 epoch) | 61.2±0.4 | 65.6±0.3 | 65.3±0.6 | 66.0±0.5 | 68.9±0.2 | 56.6±0.6 | 61.3±0.5 | 58.7±0.6 | 63.0 |
| ⋆ Token Distillation (1 epoch) | 62.8±0.5 | 67.3±0.2 | 67.6±0.3 | 67.3±0.5 | 71.0±0.2 | 56.2±1.9 | 63.1±0.2 | 61.2±0.3 | 64.6 |
| NTP (2 epoch) | 61.0 | 66.1 | 65.4 | 65.9 | 68.7 | 56.7 | 61.5 | 58.9 | 63.0 |
| ⋆ Token Distillation (2 epoch) | **63.2** | **68.5** | **68.9** | **67.7** | **71.8** | **58.4** | **63.6** | **61.3** | **65.4** |

Table 14: Comparison of different initialization methods for domain adaptation. We report a macro-average of the tasks in the Open Medical-LLM leaderboard (see Section 4.1). The best initialization result for each model is given in **boldface**. We only run a single seed for the two epoch variants.

budget – still Token Distillation outperforms ZeTT. In return, ZeTT is faster at inference time (in our experiments on a single H100 80GB GPU, ZeTT took less than a minute). In future work, we believe that investigating an Token Distillation-style objective for hyper-network pretraining would be beneficial, as our experiments show that this significantly outperforms next-token prediction for learning new embeddings (ZeTT uses next-token prediction).

# E USE OF LLMS

LLMs were used for LLM-as-a-Judge style evaluation, as discussed in the paper. LLMs were also used for formatting of Section 3 (`gpt-o4-mini-high`), intermittently while writing the code for experimentation (VSCode Copilot autocomplete), as well as for the creation of scripts that aggregate the final results into tables and figures. All LLM-generated code has been checked for correctness.

