# OpenReview forum: "Token Distillation: Attention-Aware Input Embeddings for New Tokens"
_ICLR.cc/2026/Conference — ICLR 2026 Poster_

### Official Review · Reviewer_3UEU · 2025-10-26

**Soundness:** 2
**Presentation:** 3
**Contribution:** 3
**Rating:** 4
**Confidence:** 4

**Summary:**

This paper proposes a method to train new input embedding parameters for added tokens using hidden state distillation on the last layer. Specifically, the authors align the hidden state of added tokens t* with the one of original sub-tokens $\tau (t^*)$ in related corpus and distill the hidden state with mean-squared error loss function. Experimental results on 8 decoder models show that the proposed method outperforms other vocabulary adaptation methods, and performs better with weighted next token prediction objective. Further analyses illustrate that better embedding initialization method can brings further improvement.

**Strengths:**

- It is interesting to incorporate the Transformer layer into the initialization of new input embedding parameters.
- The experimental results are promising for Transformer decoder models in the vocabulary adaptation task.

**Weaknesses:**

- Experiments are conducted on the Transformer decoder models. It is unclear for its performance on the vocabulary adaptation of Transformers encoder models.

- Missing details that the number of tokens added in the experiments. The performance of this method may be affected by the lexical similarity between the target token and source token. For example, if all target tokens like Arabic or Chinese words are significantly different to the source tokens like English words and original subtokens $t_i$ are UTF-8 characters, how does your method perform? Can your method improved by initialization method based on semantic alignment like Focus[1] and TokAlign[2] in this setting?

- Typo at the Line 359: "NTP outperforms NTP" --> "NTP outperforms NTP (tune all embeddings)"

**References**

[1] Konstantin Dobler and Gerard de Melo. FOCUS: Effective Embedding Initialization for Monolingual Specialization of Multilingual Models. EMNLP 2023.

[2] Chong Li, Jiajun Zhang, and Chengqing Zong. TokAlign: Efficient Vocabulary Adaptation via Token Alignment. ACL 2025.

**Questions:**

1) How about the results of your method on encoder Transformer models like BERT?

2) If all target tokens like Arabic or Chinese words are significantly different to the source tokens like English words and original subtokens $t_i$ are UTF-8 characters, how does your method perform? Can your method improved by initialization method based on semantic alignment like Focus and TokAlign in this setting?

---

> ### Author Response · Authors · 2025-11-21
>
> Thank you for appreciating the motivation behind our paper by noting that "[incorporating] the Transformer layer into the initialization" is "interesting" and concluding that our “experimental results are promising”. We appreciate your comments on model architecture and related work, and we will fix the typo you have pointed out. Below, we address each weakness and question in turn and clarify the experimental details.
>
> **W1/Q1: Encoder models**
>
> While we do not evaluate our method on BERT-style models, our objective in Eq. 1 is agnostic to the type of attention mask. Regarding additional experiments on BERT-style models, we kindly ask for your understanding that while more experiments are always possible, each paper needs to limit the evaluation scope. Our paper already includes extensive experiments on 9 models from at least four different families (Llama-3, Llama-3.1, Llama-3.2, Mistral, Olmo-2, Qwen3), with a wide range of baselines, ablations, and analysis experiments. These models include base and instruction-tuned variants, multiple parameter sizes (3B, 7B, 8B), and both tied and untied embedding matrices. Given the widespread use of decoder-only models, we hope this evaluation is sufficient to demonstrate significant impact.
>
> **W2: Number of added tokens**
>
> We do describe our methodology for selecting new tokens (l. 227ff and Appendix C.1/C.2 starting at l. 1008ff) and report their numbers in Table 10 (Appendix D). We add roughly 2.6k new tokens in the biomedical domain. For French, we accidentally omitted the number; it is ~11k added tokens, which we will include in the updated version.
>
> **W2/Q2: Heavily fragmented target tokens**
>
> Our objective depends on hidden states, not on the specific subtokenization. Even if a word is split into UTF-8 bytes, the model has learned a contextual representation for that sequence, which we distill into a single token. Token Distillation therefore does not rely on lexical similarity, which was one of our motivations compared to overlap-based methods like FOCUS.
>
> However, the initial starting point before Token Distillation (subtoken-mean) can indeed be weaker in heavily fragmented cases (e.g. UTF-8 bytes). Methods such as FOCUS and TokAlign, which are based on semantic alignment, could provide stronger starting points here, but this is orthogonal to our main contribution. We will add this nuance to our discussion on initial starting points from Table 6, and we thank you for highlighting the TokAlign prior work, which we will also add in our related work.
>
> Based on your question, we have also run additional experiments adding Arabic tokens to Mistral-7B, confirming that Token Distillation outperforms other baselines even when the added tokens have low lexical similarity with the source tokens (full details in [Additional Experiment C](https://openreview.net/forum?id=n20ml5nGEo&noteId=mz7mYj73nT) in our top-level comments).
>
> We thank you for the time and effort you have put into reviewing our paper and further appreciate any additional time taken for reading and reviewing our response.

---

### Official Review · Reviewer_KJ3L · 2025-10-26

**Soundness:** 3
**Presentation:** 3
**Contribution:** 2
**Rating:** 4
**Confidence:** 3

**Summary:**

This paper introduces "Token Distillation," a novel method for initializing input embeddings for new tokens in a pretrained language model. The method addresses the inefficiency and potential performance issues caused by the "over-tokenization" of domain-specific terms into multiple subtokens. The authors propose a distillation objective: to optimize a new single token's embedding such that it replicates the model's internal hidden states when processing the original sequence of subtokens. The core contribution is this efficient initialization technique that leverages the full model's knowledge, not just the embedding layer.

**Strengths:**

1. Novel and Effective: The paper proposes a clever and elegant method, "Token Distillation," which distills the model's internal behavior (hidden states) rather than simply aggregating embedding vectors. Experiments robustly show it outperforms strong baselines.
2. Thorough Experimental Validation: The claims are supported by rigorous experiments across a diverse set of models, tasks (domain and language adaptation), and a comprehensive suite of baselines, demonstrating the method's reliability.

**Weaknesses:**

1. Insufficient Justification for Practical Significance: The paper's primary weakness is the lack of compelling evidence for why the proposed efficiency-performance trade-off is critical. In many high-stakes domains, even a small performance drop is unacceptable, and the paper fails to demonstrate scenarios where the efficiency gain from token compression is a mission-critical requirement rather than a minor convenience.
2. Misaligned Motivation: There is a narrative disconnect. The motivation suggests over-tokenization harms performance, yet the experiments consistently show the original (over-tokenized) method as the performance ceiling. This weakens the argument and shifts the method's entire value proposition to an efficiency gain that is not sufficiently justified.

**Questions:**

1. My primary concern, which currently limits my rating, is about the practical necessity of the efficiency-performance trade-off this paper proposes. If the authors can convincingly address this, I am open to raising my score.
For maximum performance, my default assumption is that one would always prefer the original tokenization, accepting its computational cost. The significance of your paper hinges on proving that the efficiency gain is not just a marginal benefit but a critical enabler for certain applications. Could you provide a concrete, quantitative analysis of these efficiency benefits? For instance(Do not to do all, just example):
* Can you demonstrate how the reduced sequence length translates to significant cost savings (e.g., in dollars or GPU hours) for large-scale batch processing of a corpus?
* Can you showcase a task involving long documents where the original tokenization would force truncation and lead to failure, while your method would succeed by fitting within the context window?
* Are there real-time applications with strict latency budgets where your method is viable but the original is not?
Providing evidence for a scenario where the original tokenization is practically infeasible or prohibitively expensive would directly address this concern and strongly bolster the paper's contribution.

2. Table 3 shows a surprising case where Token Distillation outperforms the original tokenization (Llama-3-8B-Instruct on French). Could you elaborate on this? Is it a statistical anomaly, or does it suggest that for some out-of-domain languages, your distillation process might be correcting for certain suboptimalities in how the base model processes subtoken compositions?

---

> ### Author Response · Authors · 2025-11-21
>
> Thank you for appreciating our "clever and elegant" proposed method, "rigorous experiments" and "thorough experimental evaluation". We also thank you for engaging with our work in your thorough review and for your detailed suggestions. Regarding the questions and weaknesses raised, we believe we can fully address these.
>
> **Q1, W1: Practical Significance**
>
> Efficiency gains through reduced token usage are directly linked to reduced necessary FLOPs and often directly linked to business metrics (e.g. $$$).
> Think of serving models to non-English users. Here, over-tokenization is directly linked to increased computational/energy costs and significant (and expensive) efforts are made to increase efficiency (e.g. custom kernels, bespoke chips), even at the cost of task performance (e.g. via quantization). Significant speedups just by tackling the tokenization layer are very relevant here. The importance of tokenization, particularly also through the lens of equity and fairness when users have to pay "per token" – which significantly disadvantages users in over-tokenized languages, has also been discussed in prior work [1,2].
> Other critical applications for more efficient tokenization include memory-constrained settings such as on-device or increasing the amount of content coding agents such as Claude Code can retain before the context has to be pruned.
>
> Based on your suggestions, we have now added empirical measurements of computational speedups for batch processing and training on a corpus (see [*Additional Experiment C*](https://openreview.net/forum?id=n20ml5nGEo&noteId=mz7mYj73nT) in our top-level comments). Our adapted tokenizer enables significant speedups of up to 50% in wall-clock time.
> To connect this to cost, we provide a back-of-the-envelope extrapolation: a continued training run of a 7B model on the full `arb_Arab` corpus from `HuggingFaceFW/fineweb-2` (62M documents, no truncation) using the adapted tokenization rather than the original could save on the order of 6.5k USD. This estimate is based on the average tokens/document of ~7.9k under the original tokenization vs. ~5.1k under the adapted tokenization, yielding required ~11.1k H100 hours vs. ~5.7k hours based on our measurement and 1.20$/H100 hour (price from [vast.ai](https://vast.ai/pricing/gpu/H100-PCIE)).
>
> We believe this discussion and the additional results we have provided based on your suggestions will improve our work and we will include all new discussion and experiments in the updated version.
>
> **W2: Motivation**
>
> The negative consequences of overtokenization in general have been demonstrated in prior work [3,4]. However, adding new tokens to an existing model is not enough, we also need new embeddings. Here, we have shown Token Distillation to outperform prior work but do not yet match the original tokenization across all models&domains. In the paper, we argued that following established work [5,6,7], further training can improve these results beyond the original tokenization and that Token Distillation provides the best starting point for this. We have now run additional experiments with:
> 1. additional computational budget allocated to Token Distillation ([*Additional Experiment A*](https://openreview.net/forum?id=n20ml5nGEo&noteId=l5tb5qxStQ) in our top-level comments)
> 2. continued training ([*Additional Experiment B*](https://openreview.net/forum?id=n20ml5nGEo&noteId=fNXxaYuRdT) in our top-level comments)
>
> Given this increased computational budget, Token Distillation can now closely match or even outperform the original tokenization, all while offering more efficient tokenization and outperforming prior published methods.
>
> **Q2: Llama3-8B-Instruct in Table 3**
>
> In specific instances, "fixing" the tokenization of badly fragmented tokens coupled with a good initialization could be able to correct suboptimal learned processing. Token Distillation on its own (without $\alpha$NTP) optimizes to match the original model’s behavior but since this objective is never perfectly met, the solution found by Token Distillation could generalize in beneficial ways, especially since a semantic unit is now represented as a single token to the Transformer. We believe this is the case in the result reported for Llama3-8B-Instruct in Table 3.
>
> ---
> [1] Do All Languages Cost the Same? (https://arxiv.org/abs/2305.13707)
>
> [2] An Empirical Study on Cross-lingual Vocabulary Adaptation for Efficient Language Model Inference (https://arxiv.org/abs/2402.10712)
>
> [3] How Good is Your Tokenizer? (https://arxiv.org/abs/2012.15613)
>
> [4] Tokenizer Choice For LLM Training: Negligible or Crucial? (https://arxiv.org/abs/2310.08754)
>
> [5] FOCUS: Effective Embedding Initialization for Monolingual Specialization of Multilingual Models (https://arxiv.org/abs/2305.14481)
>
> [6] Zero-Shot Tokenizer Transfer (https://arxiv.org/abs/2405.07883)
>
> [7] AdaptiVocab: Enhancing LLM Efficiency in Focused Domains through Lightweight Vocabulary Adaptation (https://arxiv.org/abs/2503.19693)

---

> > ### Comment · Reviewer_KJ3L · 2025-11-23
> >
> > Thank you for your response. The additional clarifications and experiments have addressed most of my concerns. I will raise my score.

---

### Official Review · Reviewer_h7PH · 2025-10-30

**Soundness:** 2
**Presentation:** 2
**Contribution:** 2
**Rating:** 6
**Confidence:** 4

**Summary:**

This paper proposes a novel embedding initialization method for new tokens, motivated by the fact that conventional methods only consider interactions within embeddings. The proposed method instead considers the output hidden representations of the new and the corresponding base tokens and tries to make them close to each other by using an MSE loss function, while only tuning the new token representations. The experiments show that the proposed method works as the best starting point for adaptation among existing baselines, including a sophisticated method like ZeTT (which uses a hypernetwork). The analysis covers a wide range of aspects related to the design and implementation of the proposed method, showing the robustness of the method.

**Strengths:**

1. The paper is well-motivated. The paper clearly identifies the problem with conventional methods for embedding initialization of new tokens (i.e., not accounting for higher-layer model dynamics) and the necessity of training an auxiliary model for sophisticated methods like ZeTT. The proposed method clearly addresses these issues and employs a lightweight training-based approach that aims to make the representations of a new token and its corresponding source tokens similar using an MSE loss.

2. The experiments well support the advantage of the proposed method against a series of baselines, showing it can provide a good starting point for adaptation.

3. The analysis is quite extensive, covering different aspects of the proposed method, including training data, embedding initialization methods for new tokens, training objectives, and choice of a target layer.

**Weaknesses:**

1. The major limitation of this paper is a lack of full continual pre-training results. While the proposed method can provide a better starting point as “initialization”, it does not always guarantee a better performance after continual pre-training on target data. Any gains seen at the starting point might not hold up when we conduct further tuning on the target data. Given that almost all methods exhibit worse performance after embedding adaptation, they inevitably require continual pre-training before actual use. Otherwise, I do not see any point in using these methods.

2. The choice of 2,500 new tokens (L202) warrants justification. This number should substantially affect the resulting performance, both downstream and inference efficiency. The current paper does not have an analysis of different sizes of new tokens. Also, the paper lacks a description of how it generates (defines) new tokens.

3. The paper does not involve any analysis related to inference efficiency.  Given that the motivation of this line of work is to improve suboptimal tokenization, thereby achieving better downstream performance and better inference efficiency, the paper would need to demonstrate both.

**Questions:**

1. How does the proposed method ensure each training sample only contains a single new token? If there are two or more new tokens in a single sample, does Eq. (1) still work?

2. On Weakness 1, I would suggest adding a few experiments that conduct continual pre-training with baselines (including the source model) and the proposed approach, given the same training budget.

3. On Weakness 2, I would suggest incorporating an analysis of different vocabulary sizes.

4. On Weakness 3, I would suggest adding an analysis of inference efficiency. This will further strengthen the contribution of the paper if the proposed method achieves better efficiency.

---

> ### Author Response · Authors · 2025-11-21
>
> Thank you for recognizing our “novel embedding initialization method”, noting that the paper is “well-motivated” and “clearly identifies the problem” with conventional approaches, and for describing our analysis as “quite extensive” and “showing the robustness of the method”. Below, we address your main weaknesses (continual pre-training, vocabulary size, inference efficiency) and your questions.
>
> **W1/Q2: Continued pretraining experiments**
>
> Based on your suggestions, we have run experiments showing (1) with additional computational budget allocated to Token Distillation, our method is able to closely match the original tokenization (see [*Additional Experiment A*](https://openreview.net/forum?id=n20ml5nGEo&noteId=l5tb5qxStQ) in our top-level comments) and (2) if this is not the case, we show on Arabic that a moderate continued training for vocabulary adaptation does indeed boost the performance of Token Distillation to match or slightly exceed the original tokenization (see [*Additional Experiment B*](https://openreview.net/forum?id=n20ml5nGEo&noteId=fNXxaYuRdT) in our top-level comments). In all these experiments, Token Distillation remains the strongest initialization method.
>
> **W2/Q2: Choice of 2500 new tokens**
>
> We describe our methodology for selecting new tokens in l. 227ff and Appendix C.1/C.2 at l. 1008ff. We include multi-token words from our benchmark datasets, with filtering based on frequency. The idea is that adding tokens that never appear in the benchmarks would not affect downstream scores, so they would not tell us anything about the quality of the embedding initialization.
> For French, this procedure actually yields ~11k new tokens (we accidentally omitted including this number and will add it in the updated version). The main focus of our work is embedding initialization, not proposing a new vocabulary construction scheme; the token selection is chosen to maximize the sensitivity of benchmark scores to the quality of the embedding initialization. We could also do runs with fewer new tokens, but this would then simply give a weaker signal.
>
> **W3/Q4: Inference efficiency evaluation**
>
> Based on your and Reviewer `KJ3L`’s suggestions, we have added concrete efficiency experiments and explicit token-count reductions (see the [*Additional Experiment C*](https://openreview.net/forum?id=n20ml5nGEo&noteId=mz7mYj73nT) in our top-level comments). Our adapted tokenizers enable significantly faster processing (up to 50% reduced processing time) while maintaining or even improving over the original tokenization's downstream performance after moderate further training. The motivation for Token Distillation is thus both potentially improved performance after continued training and signficant efficiency improvements due to better tokenization.
>
> **Q1: What if multiple new tokens are in a single sample?**
>
> This is not a problem. During Token Distillation, we can still use the original tokenization for all but one new token. Concretely, we first tokenize with the original tokenizer -- this is $s_t$ from the paper -- and then construct $s_{t^*}$ by merging the consecutive token IDs corresponding to the new token. However, our implementation also handles multiple new tokens at the same time (we briefly discuss this in Appendix C.8 / l. 1169).

---

> > ### Comment · Reviewer_h7PH · 2025-11-25
> >
> > Thanks for the response, which addresses most of my concerns. Nonetheless, please update the paper accordingly. I will increase my score.

---

### Author Response · Authors · 2025-11-21
**Additional Experiments**

Several reviewers suggested that, to better motivate use cases for our method, we should (i) demonstrate that Token Distillation can match or exceed the original tokenization (e.g., after a continued training phase) on benchmarks, and (ii) report concrete efficiency gains such as reduced token counts and realized speedups.

In response to these suggestions, we have **added additional experiments and analysis**:
1. We added experiments showing that by **increasing the number of training samples** allocated per new token in Token Distillation by 4x (from 25 to 100), **Token Distillation** can **match the original tokenization while offering speedups** due to the better tokenization.
2. On Arabic, where Token Distillation does not initially match the original tokenization, we run a **moderate continued training phase for vocabulary adaptation**. This **lifts performance to match or slightly exceed the original tokenization**, while maintaining **substantial token-count reductions** and corresponding efficiency gains.
3. We empirically demonstrate that **reduced over-tokenization indeed translates into concrete speedups in both inference-only and training-style runs**, with wall-clock improvements of up to 50% in our measurements.

Since the **same additional experiments have been added in response to multiple reviewers**, we describe their **details in one single place in comments below** in order to reduce duplication. In the individual responses to the reviewers, we refer to these details and further clarify and specifically address questions raised by the reviewers. **We thank the reviewers for their comments and admirable engagement with our work** and believe the **framing of our proposed method has been meaningfully improved** as a result of this reviewing process.

---

> ### Author Response · Authors · 2025-11-21
> **Additional Experiment A: Token Distillation w/ more compute budget**
>
> **Additional Experiment A: Token Distillation w/ more compute budget**
>
> In the paper, we focused on a quick-to-run setting ("2500 tokens in 10 minutes"), where Token Distillation does not yet fully match the original tokenization's performance (though it significantly outperforms strong baselines inlcuding pretrained hyper-networks). Before moving to continued pretraining, we first show that increasing the budget for Token Distillation allows us to closely match the original tokenization while retaining substantial speedups, without modifying the Transformer backbone.
>
> We repeat the French vocabulary adaptation experiments from the submission, but now:
> 1. Increase the number of samples per new token from 25 to 100 (4x).
> 2. Use 5000 new tokens selected via AdaptiVocab [1] instead of the mechanism from our submission. AdaptiVocab selects frequent token-ngrams and crucially does not restrict new tokens to whole words, which allows for better sequence compression. This reduces average token counts by ~20% on the benchmark datasets (and by ~19% on web-crawled French documents from the `fra_Latn` split of `HuggingFaceFW/fineweb-2`) .
>
> The NTP tuning baseline uses the same increased data and compute budget as Token Distillation. We report the results below:
> |Methods (↓)|Llama-8B-i (Δ tokens)|Qwen3-8B (Δ tokens)|
> |-|-|-|
> |Original tokenization|72.1 (0%)|**81.7** (0%)|
> |Subtoken Mean|60.9 (**-21%**)|69.2 (**-20%**)|
> |NTP|66.5 (**-21%**)|79.1 (**-20%**)|
> |Token Distillation|**72.2** (**-21%**)|81.5 (**-20%**)|
>
> ---
> [1] AdaptiVocab: Enhancing LLM Efficiency in Focused Domains through Lightweight Vocabulary Adaptation (https://arxiv.org/abs/2503.19693)

---

> ### Author Response · Authors · 2025-11-21
> **Additional Experiment B: Token Distillation w/ continued training**
>
> **Additional Experiment B: Token Distillation w/ continued training**
>
> We next show that when Token Distillation cannot yet match the original tokenization "out-of-the-box", a moderate continued training for vocabulary adaptation phase can remedy this and even surpass the original tokenization. We report Arabic vocabulary adaptation for Mistral-7B to also address Reviewer `3UEU`’s question about low lexical similarity of new tokens to the source tokenizer (i.e. Arabic vs. mainly English). Adding the already just the top 1k tokens selected via AdaptiVocab reduces token counts on Arabic benchmarks (from the OALL Arabic Leaderboard) by 36%.
>
> Using the same increased budget as in [*Additional Experiment A*](https://openreview.net/forum?id=n20ml5nGEo&noteId=l5tb5qxStQ), Token Distillation alone does not yet match the original tokenization ("No Train" column). We then run moderate continued pretraining for vocabulary adaptation, again based on AdaptiVocab, which trains the embedding matrices as well as the first and last layers. However, we use a larger training budget than in their original paper to address the point that longer training might equalize between different initialization methods. Concretely, we use an effective batch size of 128 sequences of max length 2048 and train for 1000 steps, which amounts to ~260M training tokens. We use a learning rate of 5e-5 with 100 warmup steps, cosine decay and the AdamW optimizer.
>
> ||Train|No Train|Δ tokens|
> |-|-|-|-|
> |Original tokenization|46.9|**47.4**|0%|
> |Subtoken Mean|45.4|36.2|-36%|
> |NTP|46.1|40.6|-36%|
> |Token Distillation|**47.5**|44.2|-36%|
> |Token Distillation + $\alpha$NTP|**47.9**|45.3|-36%|
>
> After continued training, vocabulary adaptation with Token Distillation is on par with, or slightly better than, the original tokenization, which does not benefit from continued training -- likely due to its heavily over-fragmented tokenizer. At the same time, Subtoken Mean and NTP cannot match the original tokenization, although the gap to Token Distillation has gotten smaller. We also report Token Distillation + $\alpha$NTP, as we hypothesized that some NTP signal may help when new tokens are strongly OOD. Indeed, this variant performs even slightly better than "vanilla" Token Distillation. All methods with vocabulary adaptation offer signficantly reduced token counts of -36% on average.

---

> ### Author Response · Authors · 2025-11-21
> **Additional Experiment C: Computational speedups**
>
> **Additional Experiment C: Computational speedups**
>
> We finally demonstrate the concrete efficiency improvements from better tokenization. We compare batch processing of documents from the `arb_Arab` split of `HuggingFaceFW/fineweb-2` between:
> - The original Mistral-7B model with its original tokenizer, and
> - The adapted model using Token Distillation and the AdaptiVocab-selected Arabic tokens.
>
> We use a single H100 GPU and measure processing of 8192 documents per run. We truncate documents to 4k characters to avoid OOM and use a batch size of 4. We report results using just a forward pass as well as a training-style setting with a forward + backward pass.
>
> |Tokenization|Δ tokens|Compute savings (fwd)|Compute savings (fwd+bwd)| tok/sec (fwd)| tok/sec (fwd+bwd)|
> |-|-|-|-|-|-|
> |Original|–|–|–|41.8k|12.2k|
> |Adapted (Arabic)|-35%|-49.7%|-50.5%|52.1k|15.4k|
>
> The compute savings (~50%) are larger than the token-count reduction (35%), which is expected given the quadratic complexity of attention. Shorter sequences reduce attention overhead and thus also increase realized tokens/sec. For the same number of *tokens* of course, both models would have similar tokens/sec.

---

> > ### Public Comment · ~Konstantin_Dobler1 · 2026-03-03
> >
> > The computational speedup numbers reported in the comment above were measured on a B200 instead of an H100 as stated due to a scheduling mistake. For completeness, we repeat the same benchmark on an H100. We use a batch size of 1 as an H100 has less accelerator memory and keep other settings unchanged. We report both B200 and H100 results in the camera-ready version. The H100 results are:
> >
> > | Tokenization     | Δ tokens | Compute savings (fwd) | Compute savings (fwd+bwd) | tok/sec (fwd) | tok/sec (fwd+bwd) |
> > | ---------------- | -------: | --------------------: | ------------------------: | ------------: | ----------------: |
> > | Original         |        – |                     – |                         – |         27.3k |              7.4k |
> > | Adapted (Arabic) |     -35% |                -41.7% |                    -40.8% |         29.9k |              8.0k |
> >
> > Although absolute values differ, we observe significant computational speedups on both accelerators.

---

### Meta-Review · Area_Chair_QHw9 · 2026-01-08

**Summary:**

The paper suggests a new method to add new tokens to a pre-trained generative LLM, i.e. to generate embedding vectors for these new tokens. The authors point to a large literature on adding new tokens that mostly used average embeddings of sub-tokens of the new token. Here, the authors suggest to use distillation to optimize the new token embeddings so that the model output stays as close as possible to the original model output (on sub-tokens). The authors show that their method (Token Distillation) outperforms existing methods, albeit still results in slightly worse evaluation results compared to the original tokenization.

The reviewers liked the method and the clarity of the paper, but raised several repeating concerns. One concern was that Token Distillation still performs worse than the original tokenization, so it is unclear why one would ever use it. Another concern was that if the goal is the reduced inference speed, then this is not demonstrated.

**Reviewer Concerns:**

The authors performed substantial work to address the comments. They included additional experiments showing that Token Distillation can in fact match the performance of the original tokenization with further training. And they also empirically demonstrated the reduced inference speed. Two of the reviewers explicitly wrote that their concerns have been addressed.

I read the revised paper before reading the reviews and the rebuttals, and my main comment was that I had EXACTLY THE SAME concerns as the reviewers, despite the authors already addressing them! This is because all new results have been added to the Appendix and are barely mentioned in the main text. When I read the main text, I did not pay attention to those additional results, so I was left wondering about the same things as the reviewers. I think the authors should do a better job integrating these results into the main narrative of the paper, possible moving some of the new tables into the main text.

**Reviewer Scores:**

The original scores were 4/4/6. Two reviewers explicitly wrote that they have increased their scores, and indeed their concerns have been largely addressed in the revision. One reviewer who gave score 4 and did not respond, would most likely not have changed their score (the main concern was about encoder-only models, which is out of scope for this paper), but I feel their concerns are less critical for this paper. So I feel the final scores would be 4/6/8.

I therefore recommend acceptance.

---

### Decision · Program_Chairs · 2026-01-26

Accept (Poster)